# Detailed modeling of positive selection improves detection of cancer driver genes

Siming Zhao[1], Jun Liu[2], Pranav Nanga[3], Yuwen Liu[1], A. Ercument Cicek [4,5], Nicholas Knoblauch[1], Chuan He [2], Matthew Stephens [1,6] & Xin He [1]

Identifying driver genes from somatic mutations is a central problem in cancer biology. Existing methods, however, either lack explicit statistical models, or use models based on simplistic assumptions. Here, we present driverMAPS (Model-based Analysis of Positive Selection), a model-based approach to driver gene identification. This method explicitly models positive selection at the single-base level, as well as highly heterogeneous background mutational processes. In particular, the selection model captures elevated mutation rates in functionally important sites using multiple external annotations, and spatial clustering of mutations. Simulations under realistic evolutionary models demonstrate the increased power of driverMAPS over current approaches. Applying driverMAPS to TCGA data of 20 tumor types, we identified 159 new potential driver genes, including the mRNA methyl-transferase METTL3-METTL14. We experimentally validated METTL3 as a tumor suppressor gene in bladder cancer, providing support to the important role mRNA modification plays in tumorigenesis.

---

[1] Department of Human Genetics, University of Chicago, Chicago, IL 60637, USA. [2] Department of Chemistry, Department of Biochemistry and Molecular Biology, Institute for Biophysical Dynamics, Howard Hughes Medical Institute, University of Chicago, Chicago, IL 60637, USA. [3] Department of Computer Science, University of Chicago, Chicago, IL 60637, USA. [4] Computer Engineering Department, Bilkent University, Ankara 06800, Turkey. [5] Computational Biology Department, Carnegie Mellon University, Pittsburgh, PA 15213, USA. [6] Department of Statistics, University of Chicago, Chicago, IL 60637, USA. Correspondence and requests for materials should be addressed to M.S. (email: mstephens@uchicago.edu) or to X.H. (email: xinhe@uchicago.edu)

Cancer is caused by somatic mutations that confer a selective advantage to cells. Analyses of somatic mutation data from tumors can therefore help identify cancer-related ("driver") genes, and this is a major motivation for recent large-scale cancer cohort sequencing projects[1]. Indeed, such analyses have already identified hundreds of driver genes across many cancer types[1,2]. Nonetheless, many important driver genes likely remain undiscovered[3], especially in cancers with low-sample sizes. Here, we develop and apply new, more powerful, statistical methods to address this problem.

The basic idea underlying somatic mutation analyses is that genes exhibiting a high rate of somatic mutations are potential driver genes. However, mutation and repair processes are often significantly perturbed in cancer, so somatic mutations may also occur at a high rate in non-driver genes. Furthermore, somatic mutation rates vary substantially across genomic regions and across tumors. The challenge is to accurately distinguish driver genes against this complex background. Several main ideas have been used to address this challenge. One idea is to carefully model the background somatic mutation process, incorporating features that correlate with somatic mutation rate, such as replication timing[4]. Another idea is to exploit distinctive features of somatic mutations in driver genes: notably, mutations in driver genes tend to be more deleterious ("function bias"), and sometimes show a distinctive spatial pattern, tending to cluster together (e.g., in substrate-binding sites)[5]. Methods that leverage one or more of these ideas include MuSiC[6], MADGiC[7], the Oncodrive suite[8–10], and TUSON[11].

Despite this progress, most existing methods do not explicitly model the process that generates the observed somatic mutations, namely the interaction of mutational processes and selection[12]. Since tumorigenesis is an evolutionary process[13,14], explicit modeling of mutation and selection should be highly beneficial for analyzing somatic mutations in cancer[12,15–17]. Many methods described above construct a null model for non-driver genes that lacks selection, and derive test statistics to reject this null model, without explicitly modeling the alternative. Even recent evolutionarily motivated models[16,17] capture only the most basic impact of selection: differences in observed rates of non-synonymous versus synonymous mutations. Our approach, driverMAPS, is based on a much richer statistical model, which captures selection at the base-pair level, and allows the strength of selection to depend on measures of functional importance, such as conservation scores, SiFT[18], and PolyPhen[19]. In addition, we use a Hidden Markov Model to capture potential spatial clustering of somatic mutations into "hotspots". Our approach also introduces other innovative features: a detailed model of the background mutation processes, which accounts for known genomic features and variation across genes not captured by these features; and the use of a Bayesian hierarchical model to combine information across cancer types, and hence improve parameter estimates.

We demonstrate the power of our approach using both simulations and by applying it to TCGA data. Our explicit statistical models for mutation in both driver and non-driver genes allow us to perform realistic simulations to assess methods, which was largely impossible in the past. We found that current methods often fail to properly control the false discovery rate (FDR) for driver gene discovery, and among those with reasonable FDR control, driverMAPS has substantially higher power. We applied driverMAPS to TCGA exome sequencing data from 20 cancer types. The results suggest that driverMAPS is better able to detect previously known driver genes than existing methods, without excessive false positives. In addition, driverMAPS identified 159 new potential driver genes not identified by other methods. Both literature survey and extensive computational validation suggest that many of these genes are likely to be true driver genes. The novel potential driver genes included both METTL3 and METTL14, which together form a key enzyme for RNA methylation. We experimentally validated the functional relevance of somatic mutations in METTL3, providing further support for both the effectiveness of our method, and for the potential importance of RNA methylation in cancer. We believe that our methods and results will facilitate the future discovery and validation of many more driver genes from cancer sequencing data.

## Results

**A probabilistic model of positive selection on somatic mutations.** Our approach is outlined in Fig. 1. In brief, we model aggregated exonic somatic mutation counts from many tumor samples (e.g., as obtained from a normal-tumor paired sequencing cohort). Let $Y_g$ denotes the mutation count data in gene $g$. We develop models for $Y_g$ under three different hypotheses: that the gene is a "non-driver gene" ($H_0$), an "oncogene" ($H_{OG}$), or a "tumor suppressor gene" ($H_{TSG}$). Each model has two parts, a background mutation model (BMM), which models the background mutation process, and a selection mutation model (SMM), which models how selection acts on functional mutations. The rate of observed mutation at a position is the product of the background mutation rate (from BMM) and a coefficient reflecting the effect of position-specific selection (from SMM). This coefficient is related to the selection coefficient of the mutation and effective population size under a simplified population genetic model[12]: a coefficient >1 indicates positive selection, whereas <1 indicates negative selection. The BMM parameters are shared by all three hypotheses, reflecting an assumption that background mutation processes are the same for cancer driver and non-driver genes. In contrast, the SMM parameters are hypothesis specific to capture the different selection pressures in oncogenes versus tumor suppressor genes versus non-driver genes.

To estimate the hypothesis-specific SMM parameters, we use pan-cancer training sets of known oncogenes[1] ($H_{OG}$), known TSGs[1] ($H_{TSG}$), and all other genes ($H_0$). Most of the known OGs and TSGs were discovered by direct investigation, and have strong experimental support. To combine information across tumor types, we first estimate parameters separately in each tumor type, and then stabilize these estimates using Empirical Bayes shrinkage[20]. The training sets will inevitably contain some mis-classified genes. For example, the set of "all other genes" will contain some–as yet unidentified–driver genes. Although the lists of known OGs and TSGs are carefully curated, many of these genes will act in a tumor-specific manner, so, for example, a "known" TSG will not actually be a TSG in all tumor types. Such training set errors will make our hypothesis-specific parameter estimates more similar to one another than they should be, which will in turn make our model-based approach conservative in terms of identifying new driver genes. (While in principle one could try to develop a less supervised approach to mitigate these issues, this would be more complicated, and we have not attempted this here.)

Having fit these models, we use them to identify genes whose mutation data are most consistent with the driver genes models ($H_{OG}$ and $H_{TSG}$). Specifically, for each gene $g$, we measure the overall evidence for $g$ to be a driver gene by the Bayes factor (likelihood ratio), $BF_g$, defined as:

$$BF_g := 0.5 \left[ \Pr\left(Y_g | H_{OG}\right) + \Pr\left(Y_g | H_{TSG}\right) \right] / \Pr\left(Y_g | H_0\right).$$

Large values of $BF_g$ indicate strong evidence for $g$ being a driver gene, and at any given threshold we can estimate the Bayesian

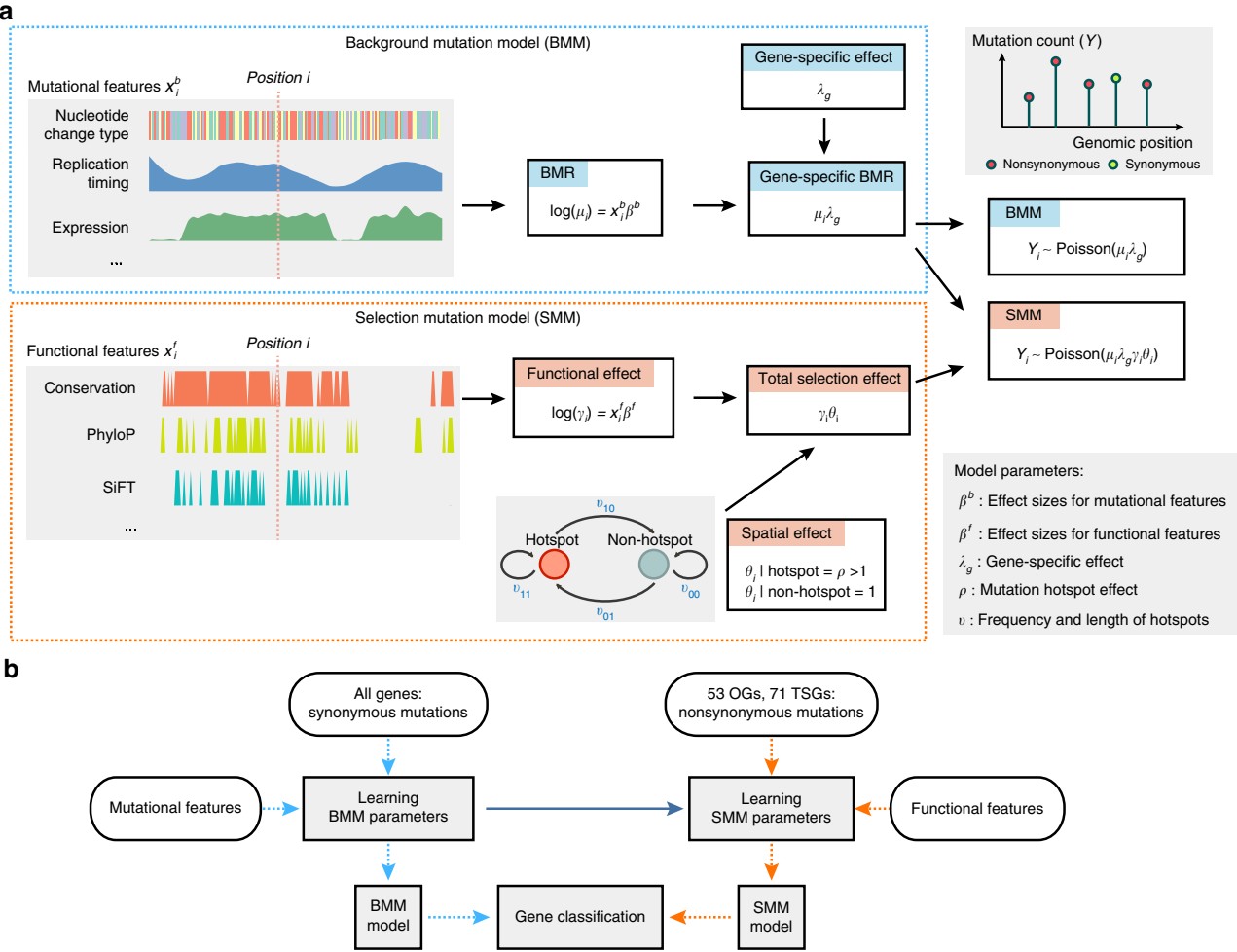

**Fig. 1** Overview of the model-based framework driverMAPS for cancer driver gene discovery. **a** Base-level Bayesian statistical modeling of mutation count data in driverMAPS. For positions without selection, the observed mutation rate is modeled by the background mutation model (BMM). Under BMM, the background mutation rate (BMR) ($\mu_i$) is determined by the log-linear model that takes into account known mutational features and further adjusted by gene-specific effect ($\lambda_g$) to get gene-specific BMR ($\mu_i\lambda_g$). For positions under selection, the observed mutation rate is modeled as gene-specific BMR adjusted by selection effect (selection mutation model, SMM). The selection effect has two components: functional effect ($\gamma_i$) takes into account functional features of the position by the log-linear model and spatial effect ($\theta_i$) takes into account the spatial pattern of mutations by the Hidden Markov Model. For both BMM and SMM, given the mutation rate, the observed mutation count data are modeled by a Poisson distribution. Note: to simplify presentation, the model here only shows mutation rate only depending on position ($i$) and not type ($t$). See the Methods section for full parameterization. **b** Gene classification workflow. Parameters in BMM are estimated using synonymous mutations from all genes. This set of parameters is fixed when inferring parameters in SMM. To infer parameters in SMM, we use non-synonymous mutations from known OGs or TSGs. driverMAPS then performs model selection by computing gene-level Bayes factors to prioritize cancer genes

FDR. For the results reported here, we chose the threshold by requiring FDR < 0.1.

**driverMAPS captures factors influencing somatic mutations.** We used a total of 734,754 somatic mutations from 20 tumor types in the TCGA project as our input data[21]. We focused on single-nucleotide somatic variations, and extensively filtered input mutation lists to ensure data quality (see the Methods section). Supplementary Fig. 1 summarizes mutation counts and cohort sizes.

The first step of our method estimates parameters of the BMM using the data on synonymous mutations. These parameters capture how mutation rates depend on various "background features" (Supplementary Table 1), which include mutation type (C > T, A > G, etc), CpG dinucleotide context, expression level, replication timing, and chromatin conformation (HiC sequencing)[4]. The signs and values of estimated parameters were generally similar across tumor types, and consistent with previous

evidence for each feature's effect on somatic mutation rate. For example, the estimated effect of the feature "expression level" was negative for almost all tumors, consistent with transcriptional coupled repair mechanisms effectively reducing mutation rate (Supplementary Fig. 2).

Our BMM also estimates gene-specific effects, using the synonymous mutations in each gene, to allow for local variation in somatic mutation rate not captured by measured features. Intuitively, the gene-specific effect adjusts a gene's estimated mutation rate downward if the gene has fewer synonymous mutations than expected based on its known features, and upward if it has more synonymous mutations than expected. A challenge here is that the small number of mutations per gene (particularly in small genes) could make these estimates inaccurate. Here, we address this using Empirical Bayes methods to improve accuracy, and avoid outlying estimates at short genes that have few potential synonymous mutations (Fig. 2a). Effectively, this adjusts a gene's rate only when the gene provides

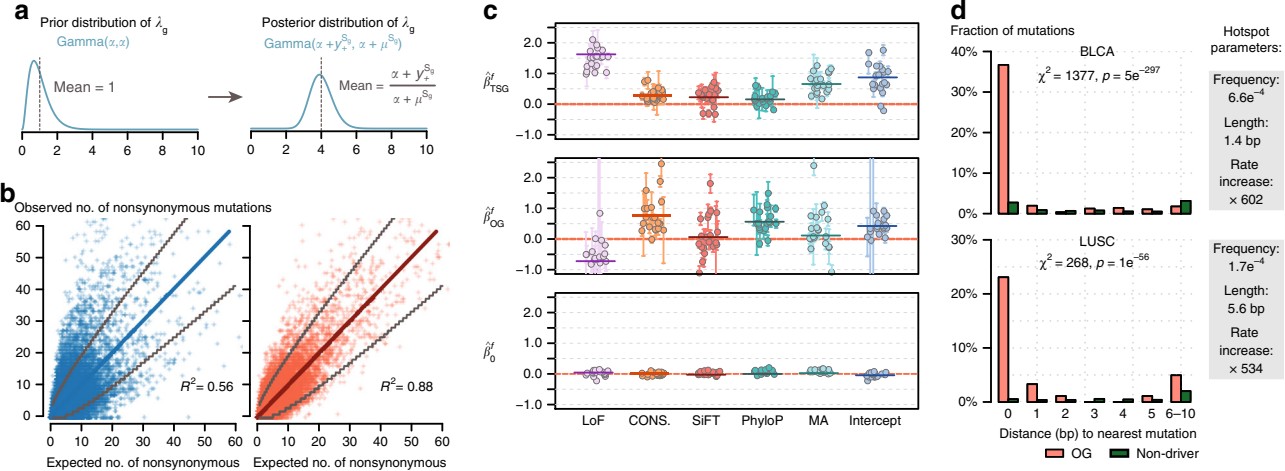

**Fig. 2** Parameter estimation results for gene-specific, functional, and spatial effects. **a** Schematic representation of how fitting synonymous mutation data affects estimation of gene-specific effect ($\lambda_g$). Note, the difference between the prior and posterior distributions of $\lambda_g$. $\alpha$ is a hyperparameter, $y_+^{S_g}$ and $\mu^{S_g}$ are the observed and expected number of synonymous mutations in gene $g$, respectively. **b** Improved fitting of observed number of non-synonymous mutations in genes with gene-specific effect adjustment. The data from tumor-type SKCM was used. The adjustment here is the posterior mean of $\lambda_g$ fitting synonymous mutation data is $(\alpha + y_+^{S_g})/(\alpha + \mu^{S_g})$. Each dot represents one gene. Gray lines indicate upper, and lower bounds of 99% confidence interval from Poisson test. The diagonal line has slope = 1, and R$^2$ was calculated using this as the regression line. **c** Effect sizes for five functional features and average increased mutation rate for TSGs (top), OGs (middle), and non-driver genes (bottom). After fitting the Background Mutation Model (BMM) using synonymous mutations, we then fix BMM parameters and used non-synonymous mutations from TSGs, OGs, and non-driver genes to fit selection models. Each dot represents an estimate from one tumor type. Horizontal bars represent mean values after shrinkage. All features are binarily coded. LoF, loss-of-function (nonsense or splice site) mutations or not. CONS., amino acid conservation; SiFT, PhyloP and MA, predictions from software SiFT[18], PhyloP[47], and MutationAssessor[48], respectively; intercept, average increased mutation rate. **d** Fraction of mutations that has the nearest mutation 0, 1, 2,.. bp away, where 0 bp means recurrent mutations. The data are shown for tumor types BLCA and LUSC. The test statistics $\chi^2$ and $p$-values were obtained in the spatial model selection procedure (see the Methods section, Supplementary Table 2). Inferred parameters related to the spatial model are shown on the right

sufficient information to do so reliably (sufficiently many potential synonymous mutations). To demonstrate the reliability of the resulting estimates, we use a procedure similar to cross-validation: we estimated each gene's gene-specific effect using its synonymous mutations, and then test the accuracy of the estimate (compared with no gene-specific effect) in predicting the number of non-synonymous mutations. (This assessment is based on an assumption that for most genes non-synonymous mutation counts will be dominated by background mutation processes rather than selection.) Fig. 2b shows the results for SKCM tumors: without the gene-specific effect, the correlation of observed and expected number of non-synonymous mutations across genes was 0.56; with gene-specific adjustment, the correlation increased to 0.88. Similar improvements were seen for other tumors (Supplementary Fig. 3).

The next step of our method estimates parameters of the SMM, using data on non-synonymous mutations. These parameters capture how the rate of non-synonymous somatic mutations depend on various "functional features" (Supplementary Data 1), including loss-of-function (LoF) status, conservation scores, etc. For non-driver genes (H$_0$), the effects of selection should be minimal, and the non-synonymous mutations should behave similarly to the BMM. This expectation is correctly reflected in the SMM parameter estimates, all of which are close to 0 (Fig. 2c, bottom panel). In contrast, under H$_{OG}$ and H$_{TSG}$ the effects of selection should be much stronger, and indeed this is correctly reflected in SMM parameter estimates that deviate substantially from 0 (Fig. 2c, top and middle panel). The SMM parameter estimates are also generally similar across tumor types, and their signs are typically consistent with expectations based on known cancer biology. For example, the estimated effect of the "LoF" feature was positive for H$_{TSG}$ and negative for H$_{OG}$, indicating that LoF mutations are enriched in TSGs and depleted in OGs, as

expected from their respective roles in cancer. The intercept terms for both TSG and OG are positive, reflecting that somatic mutations are enriched in both types of driver gene.

Although here we chose to fit our models to TSGs and OGs separately, many of the estimated effects of functional features in our SMM do not differ greatly between the two models–the primary exception is the "LoF" feature highlighted above. While the difference in estimated LoF effect makes biological sense, it also likely reflects the way that genes were assigned to be TSG versus OG in the training data, and we would caution against overinterpreting our model-based categorization of TSG versus OG based on somatic mutation data alone. Indeed, the line between TSG and OG can be blurred, with some genes acting as both TSGs and OGs in different contexts[22]. These observations raise the question of whether accuracy for detecting driver genes might be improved by pooling TSGs and OGs into a single model–instead of treating them separately as we do here–thereby reducing the number of parameters to be estimated. In simulation experiments, we found accuracy of the single-model approach to be almost identical to that obtained by using separate models (Supplementary Fig. 4). However, as training sets of TSGs and OGs improve (e.g., larger and more tumor specific), and with the incorporation of additional functional features, there may be increased benefit to separately modeling TSGs versus OGs. We therefore focus on the results from this approach for the remainder of the paper.

**driverMAPS improves detection of driver genes in simulations.** While many methods have been developed for driver gene identification, it is difficult to compare them on real data where the true status of each gene is often unknown. Simulations are extremely valuable in such situations, and have been used in

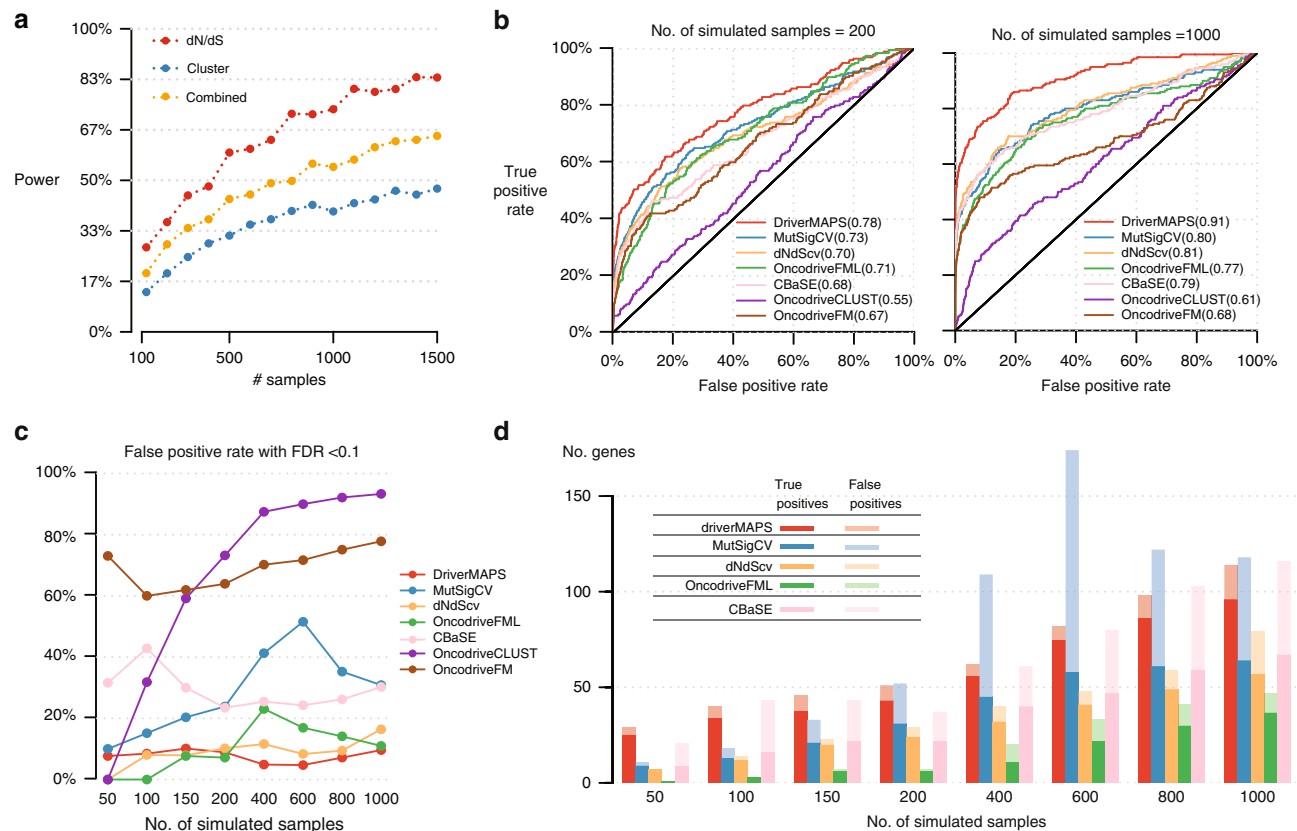

**Fig. 3** driverMAPS predicts driver genes with high accuracy and increased power in simulations. **a** Combining *p*-values from methods that use only one feature of positive selection at a time can lose power. We simulated mutations in a gene under positive selection at various sample sizes, and then assessed the power to detect this gene as positively selected. "dN/dS" captures the excess of non-synonymous mutations, "cluster" captures spatial clustering pattern of mutation, "combined" combines *p*-values from "dN/dS" and "cluster" using Fisher's method. See the Methods section for details. **b** Receiver-operating characteristic (ROC) curves of several methods applied to genome-wide simulation data. Overall, 324 genes are chosen to be positively selected (191 TSGs and 133 OGs), and the rest of genes are neutral. We used 124 out of the 324 genes as training set for driverMAPS, and used the remaining 200 genes as the test set to generate ROC curves. Area under the ROC Curve (AUROC) values are shown in parentheses. **c** False-positive rate at FDR cutoff 0.1 on the simulated data. **d** The number of true-positive and false-positive genes at FDR < 0.1. To ensure a fair comparison, this excludes the 124 training genes used to train driverMAPS

many fields, including population genetics[23], statistical genetics[24], and single-cell transcriptomics[25]. Here, we exploit our explicit statistical model to perform realistic simulations based on parameters inferred from real data (here, the TCGA LUSC cohort).

We begin by using a simple simulation to briefly motivate why a model-based approach to integrate multiple selection-related features may be preferable to an alternative strategy that is commonly used in the field: performing a hypothesis test for each of several selection-related features and then combining their *p*-values (e.g., using Fisher's method[26]). Although combining *p*-values is simple, and can be effective in some statistical settings, it also has its limitations, and we believe it is not well suited to this particular setting. To illustrate this, we simulated somatic mutations in a positively selected gene with both increased non-synonymous mutation rates and mutational hotspots. We obtained *p*-values from two simple tests—a dN/dS test to detect enrichment of functional mutations and another to detect spatial clustering (see the Methods section)—and combine them using Fisher's method. Perhaps unexpectedly, the combined test has lower power than the dN/dS test alone (Fig. 3a). We believe that this is because spatial clustering is a relatively weak feature in our simulations (as in real data), and so the spatial test has much less power than the dN/dS test. Consequently, the spatial test adds more noise than signal, decreasing power. This highlights a general weakness of methods based on combining *p*-values, that it

is difficult to take account of differences in informativeness among tests; in contrast model-based approaches like ours automatically weight different features based on their informativeness.

We next used more comprehensive simulations to compare driverMAPS with six existing algorithms: MutSigCV, OncodriveFML[9], OncodriveFM[10], OncodriveCLUST[8], dNdScv[16], and CBaSE[17]. We simulate mutations in driver and non-driver genes under models that are based on the driverMAPS modeling approach, but with several modifications so that driverMAPS has to deal with realistic levels of model misspecification. In particular (i) we simulated background mutations under the mutation model of dNdScv, which uses 192 mutation types considering tri-nucleotide contexts (instead of the nine types we use in driverMAPS); and (ii) we simulated additional variation in the strength of selection at each position that is not explained by observed functional features. In addition, to add further model misspecification, when running driverMAPS, we provided it with only a subset of functional features used in simulations (see the Methods section). We simulated data for all genes in the genome, with 324 genes randomly chosen to be oncogenes or tumor suppressor genes. We found that, for distinguishing driver versus non-driver genes, driverMAPS outperformed all other methods (Fig. 3b). In terms of FDR control, only driverMAPS consistently maintains proper FDR levels across all sample sizes, with dNdScv

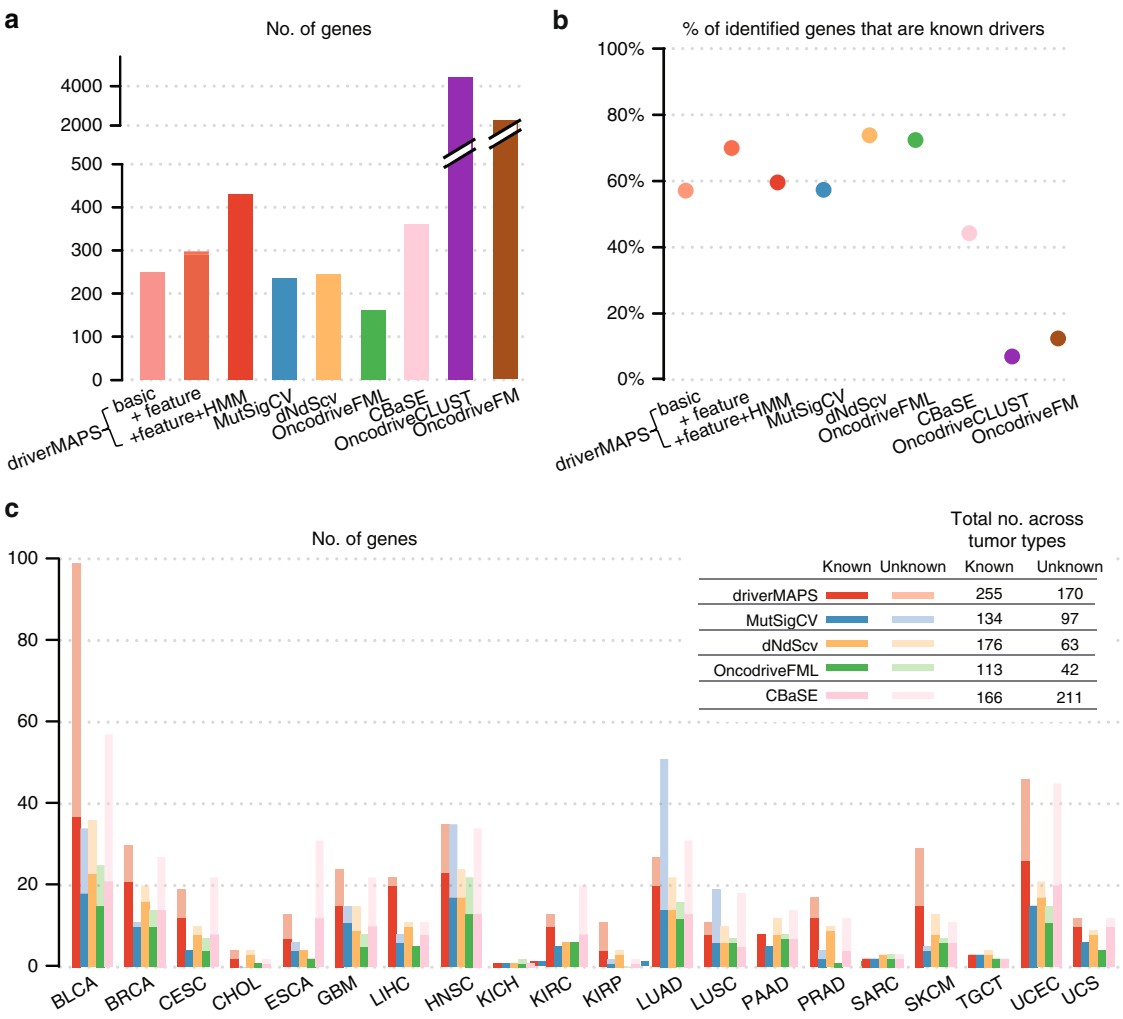

**Fig. 4** Gene prediction using TCGA somatic mutation data. **a** The total number of predicted driver genes aggregating across all cancer types. driverMAPS (Basic): driverMAPS with no functional features information and no modeling of spatial pattern; driverMAPS (+ feature): driverMAPS with all five functional features in Fig. 2, but no modeling of spatial pattern; driverMAPS (+ feature + HMM): complete version of driverMAPS with all five functional features and spatial pattern. **b** Percentage of known cancer genes among predicted driver genes aggregating across all cancer types. **c** The number of significant genes at FDR < 0.1 stratified by tumor type. For all "unknown" genes included here, we verified mutations by visual inspection of aligned reads using files from Genomic Data Commons (see Supplementary Note 6). Total numbers of known and unknown significant genes aggregating across all cancer types are summarized top right

being the second (Fig. 3c). Excluding two methods with obvious problems of FDR control (OncodriveFM, OncodriveCLUST), driverMAPS identifies the most driver genes at FDR < 0.1 (Fig. 3d). Overall, we found the power of driverMAPS to discover novel driver genes can double that of other leading methods (and even more in smaller samples).

**Application of driverMAPS on TCGA data**. We next compared the results from driverMAPS and other algorithms for predicting driver gene using the TCGA data (see Methods). Besides the full implementation of driverMAPS, we also tried a "basic" version that looks only for an excess of non-synonymous somatic mutations (without any functional features or spatial model), and a " + feature" version with functional features, but not the spatial model. We applied all methods to the same somatic mutation data, and compared the genes they identified with a list of "known driver genes" (713 genes) compiled as the union of COSMIC CGC list (version 76)[27], Pan-Cancer project driver gene list[2], and list from Vogelstein B (2013)[1] (see Supplementary Note 6). To avoid overfitting of driverMAPS to the training data, we trained

driverMAPS with a leave-one-gene-out strategy in these assessments.

For each method, we computed both the total number of genes detected (at FDR = 0.1) (Fig. 4a) and the "precision"–the fraction that are on the list of known driver genes (Fig. 4b). All versions of driverMAPS identified more driver genes than either MutSigCV, dNdScv, or OncodriveFML, while maintaining a similarly high precision. The full version of driverMAPS (with the spatial and functional features) identified nearly twice as many genes as any of these method. Furthermore, this higher detection rate of driverMAPS was consistent across tumor types (Fig. 4c). CBaSE identified the second most genes (excluding OncodriveFM and OncodriveCLUST), but the fraction of known driver genes is considerably lower than all other methods except OncodriveFM and OncodrivCLUST (see below), suggesting higher false-positive rates, as we observed in simulations (Fig. 3c). The other two methods, OncodriveFM and OncodriveCLUST, behaved quite differently, identifying thousands of driver genes but with much lower precision, consistent with poor FDR control in simulations (Fig. 3c). For OncodriveFM and OncodriveCLUST, the lowest

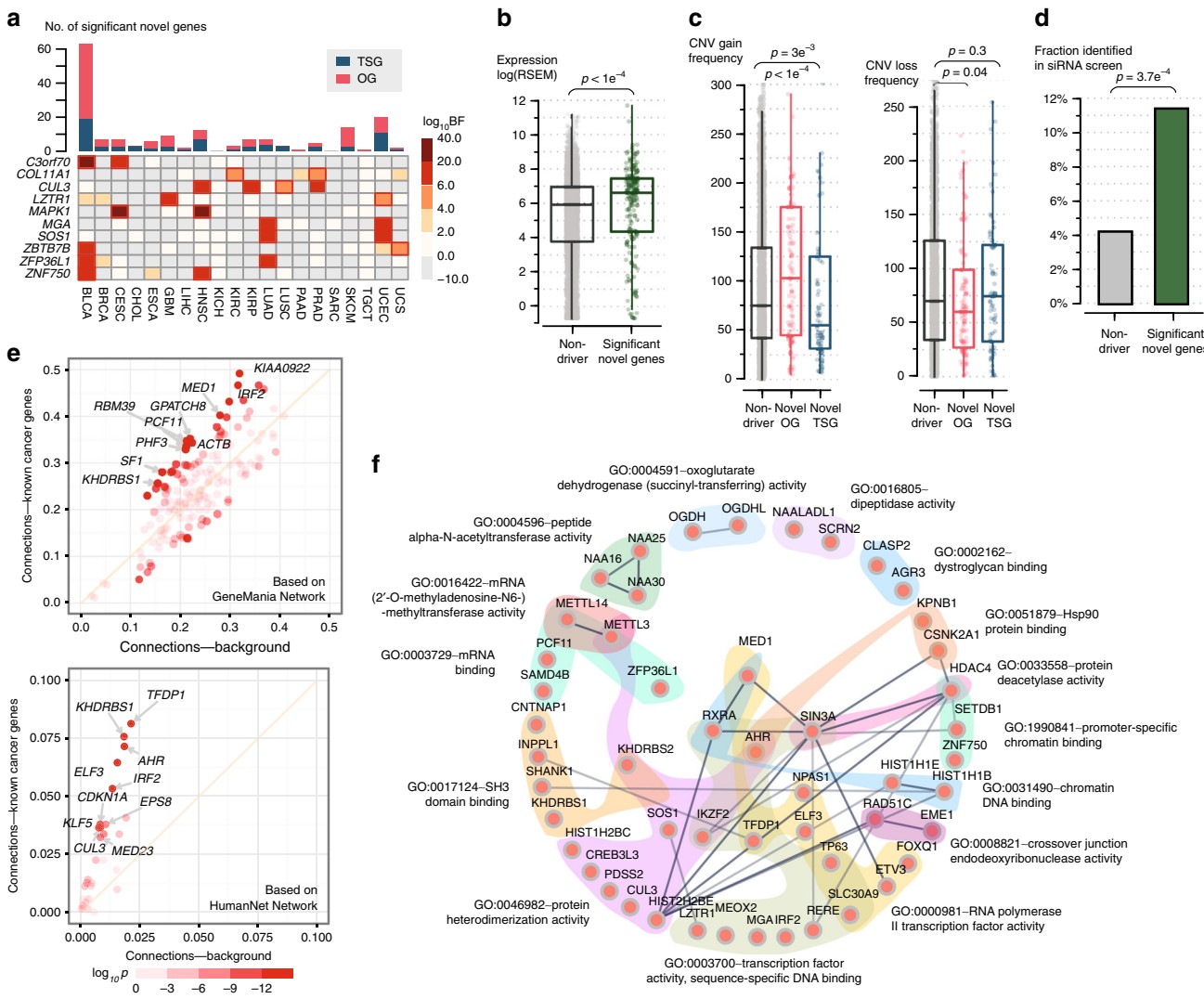

**Fig. 5** Evaluation of novel cancer genes predicted by driverMAPS. **a** Overview of predicted novel cancer genes. Top, the number of novel genes in each cancer type. Bottom, heatmap of Bayes factors (BF) for recurrent novel genes across tumor types. Significant BFs are highlighted by red boxes. **b**–**d** Predicted novel cancer genes show known cancer gene features. For each feature, quantification of the feature level in the novel cancer gene set was compared with the non-driver (neither known or predicted) gene set. The features are gene expression levels[21] stratified by tumor types the novel genes were identified from (**b**), similarly stratified copy number gain/loss frequencies[21] (**c**) and fraction of genes identified in a siRNA screen study[28] (**d**). In panels **b** and **c**, the center line, median; box limits, upper and lower quartiles. **e** Enriched connectivity of a predicted gene with 713 known cancer genes (Y-axis) compared with with all genes ($n = 19,512$, X-axis). Connectivity of a selected gene with a gene set is defined as the number of connections between the gene and gene set found in a network database divided by the size of the gene set. Each dot represents one of the 159 novel genes with ten most enriched ones labeled. Color of dots indicates two-sided Fisher exact p-value for enrichment. **f** Significantly enriched GO-term gene sets (FDR < 0.1, "molecular function" domain) in predicted novel cancer genes. GO-term[31, 32] gene sets are indicated by distinct background colors. Links among genes represent interactions based on STRING network database[49], with darker color indicating stronger evidence

precision was in the tumor types with the highest mutation rates (e.g., BLCA, LUSC, LUAD), suggesting that these methods may be adversely affected by mutation rate variation (Supplementary Fig. 5). While the precision of OncodriveFM and Oncodrive-CLUST were negatively correlated with mutation rate (Pearson $r = -0.44$ and $-0.56$), the precision of driverMAPS showed negligible correlation (Pearson $r = 0.05$).

**Evaluation of potential novel drivers identified by driverMAPS.** Summing across all 20 tumor types, at FDR 0.1, driverMAPS identified 255 known driver genes and 170 putatively novel driver genes (159 unique genes across the 20 tumor types; 70 classified as TSGs and 100 as OGs; Fig. 5a; Supplementary Table 3). Almost half of these putative novel genes were not called by MutSigCV,

OncodriveFML, dNdScv, or CBaSE. Ten novel genes were found independently in at least two tumor types (Table 1). This is unlikely to happen by chance under the null (permutation test, $p < 1e^{-4}$), so these genes seem especially good candidates for being genuine driver genes.

Since it is impractical to functionally validate all 170 putative novel genes, we sought other data to support these genes likely being involved in cancer. We first selected three common tumor types–the breast, lung, and prostate–and conducted an extensive literature survey for each novel gene identified in these tumor types. Among a total of 22 novel genes, we found clear support in the literature for 20 being involved with cancer biology, either directly implicated as oncogenes or tumor suppressor genes (but not in the list of "known driver genes") or linked to well-established cancer pathways (Supplementary Data 2).

**Table 1 Novel significant drivers found in at least two tissue types**

| Gene | #Missense | #LoF | #Silent | $\log_{10}$BF | Tumor | Function |
|---|---|---|---|---|---|---|
| C3orf70 | 14/3 | 1/1 | 0/0 | 9.3/3.8 | BLCA/CESC | Unknown |
| COL11A1 | 7/13 | 4/2 | 0/0 | 2.2/2.2 | KIRC/PRAD | Collagen formation, expression associated with colorectal, ovarian cancers, etc. (23934190, 11375892) |
| CUL3 | 15/8/4 | 5/4/0 | 1/0/0 | 3.5/3.8/2.6 | HNSC/KIRP/PRAD | Core component of E3 ubiquitin ligase complex, with many downstream targets affecting carcinogenesis, like NRF2 (24142871) |
| LZTR1 | 9/10 | 0/1 | 0/2 | 2.9/2.1 | GBM/UCEC | Adaptor of CUL3-containing E3 ligase complexes. Inactivation drives glioma self renewal and growth (23917401) |
| MAPK1 | 9/7 | 0/1 | 0/0 | 15.1/ 12.8 | CESC/HNSC | MAP kinase. The MAPK/ERK cascade has well characterized and important roles in cancer (17496922) |
| MGA | 35/11 | 16/5 | 5/3 | 3.8/2.7 | LUAD/UCEC | Dual-specificity transcription factor, can inhibit MYC-dependent cell transformation (10601024) |
| SOS1 | 12/7 | 1/0 | 3/0 | 3.5/7.0 | LUAD/UCEC | Guanine nucleotide exchange factor for RAS proteins, which are well-known for roles in cell proliferation (17486115) |
| ZBTB7B | 11/5 | 1/1 | 0/0 | 6.2/2.3 | BLCA/UCS | Transcriptional regulator of lineage commitment of immature T-cell precursors (17878336) |
| ZFP36L1 | 12/11 | 4/3 | 1/0 | 3.4/5.2 | BLCA/LUAD | Involved in mRNA degradation. Deletion leads to T lymphoblastic leukemia (20622884) |
| ZNF750 | 17/13 | 3/7 | 2/1 | 3.4/5.1 | BLCA/HNSC | An essential regulator of epidermal differentiation. Depletion promotes cell proliferation in ESCA (24686850) |

We use "/" to separate data obtained from different tumor types as indicated in the "Tumor" column. A brief description of the gene's function and its known role in cancer is provided in the "Function" column. Reference PMIDs are given in parentheses

We next assessed whether the novel genes were enriched for features often associated with driver genes. Previous studies reported that driver genes tend to be highly expressed[4] compared with other genes, and indeed we found that, collectively, the novel genes showed significantly higher expression than randomly sampled genes in the corresponding tissues[21] (permutation test, $p < 1e^{-4}$) (Fig. 5b).

Previous studies have also reported that driver genes tend to show enrichment and depletion for different copy-number-variation (CNV) events, depending on their role in cancer. Specifically, OGs are enriched for CNV gains and depleted for CNV loss, whereas TSGs show enrichment for loss and depletion for gains. Consistent with this, we found novel genes identified as OGs are enriched for CNV gain events (permutation test, $p < 1e^{-4}$), while novel TSGs are depleted (permutation test, $p = 3e^{-3}$). CNV loss events for novel OGs are depleted compared with novel TSGs and to other genes (permutation test, $p = 0.04$) (Fig. 5c).

We also compared our novel genes with a "cancer dependency map" of 769 genes identified from a large-scale RNAi screening study across 501 human cancer cell lines[28]. These are genes whose knockdown affects cell growth differently across cancer cell lines, thus likely representing genes that are critical for tumorigenesis, but not universally essential genes. We found 16 novel driver genes overlapped with this gene list, a significant enrichment compared with random sampling (odds ratio 2.9, $p = 3.7e^{-4}$) (Fig. 5d; Supplementary Data 3).

To test whether our novel genes are functionally related to known cancer driver genes, we examined the connectivity of these two sets of genes in the HumanNet[29] gene network, which is built from multiple data sources, including protein–protein interactions and gene co-expression. On average, each novel gene is connected to 3.8 known driver genes, significantly higher than expected by chance (permutation test, $p = 0.001$). We obtained a similarly significant result using a different gene network, GeneMania[30], which is constructed primarily from co-expression (permutation test, $p = 0.008$) (Fig. 5e).

Finally, we identified enriched functional categories in our novel genes using GO enrichment[31,32] analysis (in geneSCF[33]). Significant GO terms (FDR < 0.1, Fig. 5f) include many molecular processes directly implicated in cancer, such as transcription

initiation and regulation. The significant terms also include several that have not been previously implicated in cancer. Genes NAA25, NAA16, and NAA30 (GO: 0004596) are peptide N-terminal amino acid acetyltransferases[34]. NATs are dysregulated in many types of cancer, and knockdown of the NatC complex (NAA12-NAA30) leads to p53-dependent apoptosis in colon and uterine cell lines[35]. OGDH and OGDHL (GO:0004591) have oxoglutarate dehydrogenase activities and part of the tricarboxylic acid (TCA) cycle[36]. METTL3 and METTL14 (GO: 0016422) form the heterodimer N6-methyltransferase complex, and are responsible for methylation of mRNA ($m^6A$ modification)[37]. This form of RNA modification may influence RNA stability, export, and translation, and has been shown to be important for important biological processes, such as stem cell differentiation. Our results suggest that this RNA methylation pathway may also play a key role in tumorigenesis, and so we examined the results for these genes in more detail.

**METTL3 is a potential TSG in bladder cancer.** driverMAPS identified the genes METTL3 and METTL14 as driver genes in the cohorts BLCA (bladder cancer) and UCEC (uterine cancer), respectively. These two genes had relatively low mutation frequencies (4 and 2%), and were not detected by MutSigCV, dNdScv, OncodriveFML, or CBaSE (the methods with reasonable FDR control). Inspecting the mutations in these two genes, we found many to be "functional" as predicted by annotations, and showed spatial clustering patterns in the MTase domain (Fig. 6a; Supplementary Table 4). Furthermore, METTL3 contained a single synonymous mutation, and METTL14 contained none, suggesting low baseline mutation rates at the two genes. While this paper was in preparation, METTL14 was independently identified as a novel TSG in endometrial cancer[38]. We thus focused on METTL3 in bladder cancer.

To gain further insights into the potential impact of the somatic mutations in METTL3, we performed structural analysis. By mapping mutations in the MTase domain of METTL3 to its crystal structure[39], we found them to be concentrated in two regions: one close to the binding site of S-adenosyl methionine (AdoMet, donor of the methyl group), and the other in the putative RNA-binding groove at the interface

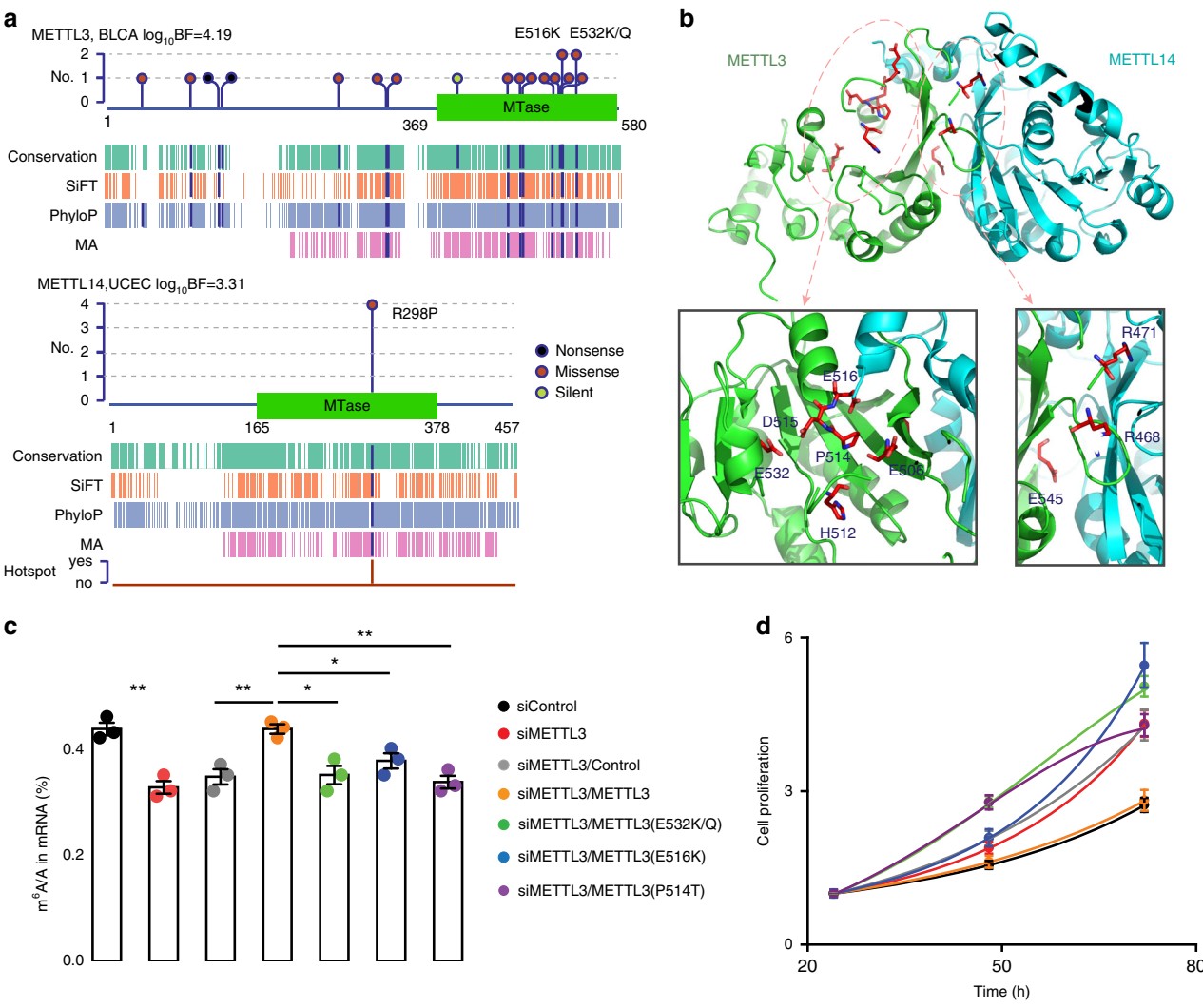

**Fig. 6** Functional validation of METTL3 as a TSG in bladder cancer. **a** Features of mutations in METTL3 and its heterodimerization partner METTL14. We show schematic representations of protein domain information and mark mutation positions by "lollipops". Recurrent mutations are labeled above. Start and end of domain residues are labeled below. Dark blue bars in aligned annotation tracks indicate the mutation is predicted as "functional". Track "Hotspot" is the indicator of whether the mutation is in hotspot or not in the driverMAPS's spatial effect model (see Supplementary Note 3). **b** Structural context of METTL3 mutations reveal two regional clusters. Top, structure of METTL3 (residues 369–570) and METTL14 (residues 117–402) complex (PDB ID: 5IL0) with mutated residues in stick presentation. Bottom, zoom-in views of the two regions with mutated residues labeled. **c** Impaired m6A RNA methyltransferase activity of mutant METTL3 in bladder cancer cell line "5637". LC-MS/MS quantification of the m6A/A ratio in polyA-RNA in METTL3 or control knockdown cells, rescued by overexpression of wild-type or mutant METTL3 is shown. **d** Mutant METTL3 decreased proliferation of "5637" cells. Proliferation of METTL3 or control knockdown cells, rescued by overexpression of wild-type or mutant METTL3 in MTS assays is shown. Cell proliferation is calculated as the MTS signal at the tested time point normalized to the MTS signal ~24 h after cell seeding. For all experiments in (**c**, **d**), the number of biological replicates is three and error bars indicate mean ± s.e.m. *$p < 0.05$; **$p < 0.01$ by two-sided $t$ test. Legend is shared between (**c**) and (**d**)

between METTL3 and METTL14 (Fig. 6b). The region close to the AdoMet-binding site contains seven mutations: E532K, E532Q, E516K, D515Y, P514T, H512Q, and E506K. Position E532 has been reported to form direct water-mediated interactions with AdoMet[39]. The other mutations map to gate loop 2 (E506K and E516K map to the start and end; the other three mutations are inside the loop) which is known to undergo significant conformational change before and after AdoMet binding. Thus all these mutations are good candidates for affecting adenosine recognition. The second region, in the METTL3-METTL14 interface, contains mutations R471H, R468Q, and E454K, and so these mutations are good candidates for disrupting METTL3-METTL14 interaction. In further support of this, the highly recurrent R298P mutation in METTL14 lies in the binding groove of the METTL14 gene.

We performed functional experiments to test whether mutations ($n = 7$) in the first region affect METTL3 function. In an in vitro assay, most mutations reduced methyltransferase activity of METTL3 (Supplementary Fig. 6, see the Methods section), and we chose four mutations (at three positions) for further cell line experiments. In two bladder cell lines ("5637" and "T24"), knockdown of METTL3 by siRNA significantly reduced m6A methyltransferase activity (Fig. 6c for "5637"; Supplementary Fig. 7a for "T24"). When we tried to rescue this phenotype by transfection of METTL3 mutants, all of the mutations, E532K/Q, E516K, and P514T failed to restore methyltransferase activity to original levels (Fig. 6c; Supplementary Fig. 7a), suggesting that they are LoF mutations.

We next examined whether disruption of METTL3 is associated with tumor progression. Indeed, knockdown of

METTL3 significantly increased cell proliferation. Wild-type METTL3 successfully restored the cells to their normal growth rate, but none of the mutants could (Fig. 6d; Supplementary Fig. 7b).

These results show that somatic mutations in METTL3 may promote cancer cell growth by disrupting the RNA methylation process, and invite further characterization of the role of METTL3 and RNA methylation in tumorigenesis by in vivo experiments.

## Discussion

We have developed an integrated statistical model-based method, driverMAPS, to identify driver genes from patterns of somatic mutation. By applying this method to data from multiple tumor types from TCGA, we detected 159 novel potential driver genes. We experimentally validated the function of mutations in one gene, METTL3. The remaining genes (Table 1; Supplementary Data 2, 3) are enriched for many biological features relevant to cancer, and appear promising candidates for further investigation.

Compared with previous methods for detecting driver genes, a key feature of driverMAPS is that it models mutation rates at the base-pair level. This allows us to explicitly model how selection strength varies based on site-level functional annotations, e.g., conservation and LoF status. This model-based approach can be thought of as a powerful extension of methods that detect driver genes by testing for an excess of non-synonymous versus synonymous somatic mutations (Nik-Zainal et al.[40], Martincorena et al.[16]), similar to the dN/dS test in comparative genomics. Indeed, the stripped-down version of driverMAPS that uses no functional annotation or spatial model is conceptually a dN/dS test (driverMAPS-basic in Fig. 4). The full version of driverMAPS, by incorporating additional functional annotations and spatial modeling, allows that some non-synonymous mutations may be more informative than others in identifying driver genes. Furthermore, by estimating parameters in a single-integrated model, our approach learns how to weigh and combine the many different sources of information. The results in Figs 3 and 4 demonstrate the increased power that comes from these extensions.

Our statistical and experimental results for the mRNA methyltransferase METTL3 add to the growing evidence of links between mRNA methylation and cancer. Indeed, a recent study, in myeloid leukemia cell lines[41], found that depletion of METTL3 also leads to a cancer-related phenotype. Extensive functional studies of METTL14 in uterine cancer[38] support a role for this gene in cancer etiology. However, intriguingly, our results on METTL3 in bladder cancer, and the METTL14 results in uterine cancer suggest that they act as tumor suppressor genes, whereas the data on METTL3 in myeloid leukemia cell lines are more consistent with an oncogenic role, with depletion inducing cell differentiation and apoptosis[41]. Further studies in multiple tumor types therefore seem necessary to properly characterize the role of mRNA methylation in cancer.

Although our model incorporates many features not considered by existing methods, it would likely benefit from incorporating still more features. For example, it may be useful to incorporate data on protein structure, which affects the functional importance of amino acid residues. Furthermore, whereas we currently use the same mutation model for all individuals, it could be helpful to incorporate individual-specific effects, such as smoking-induced mutational signatures[42]. Finally, it could be useful to extend the model to incorporate information on non-coding variation, which has been shown to be important for many human diseases, including cancer. Although identifying functional non-coding variation remains a major general challenge, extending our model to incorporate features from studies of epigenetic factors, such as methylation or open chromatin, has the potential to detect novel driver genes affected by non-coding somatic mutations.

## Methods

**Data preparation.** We downloaded somatic single-nucleotide mutations identified in whole-exome sequencing (WES) studies for 20 tumor types from TCGA GDAC Firehose (https://gdac.broadinstitute.org/). We obtained the MAF files using firehose_get (version 0.4.6) (https://confluence.broadinstitute.org/display/GDAC/Download) and extracted position and nucleotide change information for all single-nucleotide somatic mutations. See Supplementary Note 1 for the 20 tumor types and abbreviations.

We excluded mutations from hypermutated tumors, as they likely reflect distinct underlying mutational processes. We also performed extensive filtering to exclude likely false-positive mutations. For each tumor type, we then generated a mutation count file that contains mutation counts (aggregated across all individuals in the tumor cohort) of all possible mutations at all sufficiently sequenced positions (see Supplementary Note 1). For a tumor type with 30 million bases sequenced, this produces 90 million possible mutations in the mutation count file (since each nucleotide can mutate to three other nucleotides). The majority of counts for these possible mutations are 0 s.

For each possible mutation, we annotated it with type and gene information, mutational features and functional features. We defined nine mutation types based on nucleotide change type (such as A > T, G > A, etc.) and genomic context (such as if inside CpG) (see Supplementary Note 2 for definitions). We categorized mutations as synonymous (S) or non-synonymous (NS), as described in the "Model description" section below. The mutational features we used include gene expression, replication timing, and HiC sequencing downloaded from http://archive.broadinstitute.org/cancer/cga/mutsig. We selected five functional features describing mutation impact. See Supplementary Note 2 for feature details. The features were added to the mutation count file by ANNOVAR[43].

**Model description.** We model each tumor type separately, so here we describe the model for a single tumor type. Let $Y_{it}$ denotes the number of mutations of type $t$ (defined by base substitution) at sequenced position $i$, across all samples in a cohort. Let NS denotes the set of non-synonymous mutations. That is, NS is the set of pairs $(i,t)$ such that a mutation of type $t$ at sequence position $i$ would be non-synonymous. (We also include synonymous mutation with a high splicing impact score in NS; see Supplementary Note 3.) Similarly, let S denotes the remaining (synonymous) $(i,t)$ pairs.

*BMM.* For synonymous mutations, we assume the following "background mutation model":

$$Y_{it}|H_m \sim \text{Poisson}\left(\mu_{it}\lambda_{g(i)}\right) \text{ [for } (i,t) \in \text{S]},\qquad(1)$$

where $\mu_{it}$ represents a background mutation rate (BMR) for mutation type $t$ at position $i$, and $\lambda_{g(i)}$ represents a gene-specific effect for the gene $g(i)$ that contains sequence position $i$. Note that the parameters of this BMM do not depend on the model $m$, so $P(Y^{S_g}|H_m)$ is the same for all $m$.

We allow the BMRs to depend on mutational features (e.g., the expression level of the gene) using a log-linear model:

$$\log\mu_{it} = \beta_{0t}^b + \sum_j x_{ij}^b\beta_j^b,\qquad(2)$$

where $x_{ij}^b$ denotes the $j$th background feature of position $i$ (not dependent on mutation type), $\beta_{0t}^b$ controls the baseline mutation rate of type $t$, and $\beta_j^b$ is the coefficient of the $j$th feature. The values $x_{ij}^b$ are observed, and the parameters $\beta^b$ are to be estimated. To indicate the dependence of $\mu_{it}$ on parameters $\beta^b$, we write $\mu_{it}(\beta^b)$.

We assume that the gene-specific effects $\lambda_g$ have a Gamma distribution across genes:

$$\lambda_g \sim \text{Gamma}(\alpha, \alpha),\qquad(3)$$

where $\alpha$ is a hyperparameter to be estimated.

*SMM.* For non-synonymous mutations, we introduce additional model-specific parameters: $\gamma_{it}^m$ representing a selection effect (SE) for mutation type $t$ at position $i$ under model $m$ and $\theta_i^m$ representing a spatial effect for position $i$ under model $m$:

$$Y_{it}|H_m \sim \text{Poisson}\left(\mu_{it}\lambda_{g(i)}\gamma_{it}^m\theta_i^m\right) \text{ [for } (i,t) \in \text{NS]}.\qquad(4)$$

For all models, $m = $ OG, TSG, and the null model, we allow the selection effect to depend on functional features (e.g., the assessed deleteriousness of the mutation), using a log-linear model:

$$\log\gamma_{it}^m = \beta_0^{f,m} + \sum_j x_{ijt}^f\beta_j^{f,m},\qquad(5)$$

where $x_{ijt}^f$ denotes the $j$th functional feature of position $i$ (this depends on mutation

type; e.g., at the same position, some mutations may be more deleterious than others), $\beta_j^{f,m}$ is the coefficient of the $j$th functional feature, and the intercept $\beta_0^{f,m}$ captures the overall change of mutation rate at NS sites regardless of functional impact. To indicate the dependence of $\gamma_{it}^m$ on parameters $\beta^{f,m}$, we write $\gamma_{it}(\beta^{f,m})$.

To model the spatial effects, we use a Hidden Markov Model (HMM) with parameters $\Theta^m$,

$$\theta^m \sim f_{\mathrm{HMM}}(\cdot; \Theta^m), \tag{6}$$

In brief, this HMM allows for the presence of mutation "hotspots"–contiguous basepairs with a higher rate of mutation–and the parameters include the average hotspot length and intensity $\rho$. See Supplementary Note 3 for details.

**Parameter estimation.** *BMM.* To simplify inference, we took a sequential approach to parameter estimation. First, we infer parameters $\beta^b, \alpha$ of the BMM using the synonymous mutation data at all genes. Let $S_g$ denotes the subset of synonymous mutations S in gene $g$, and $Y^{S_g}$ denotes the corresponding observed counts:

$$Y^{S_g} = \left\{ Y_{it} : (i, t) \in S_g \right\}. \tag{7}$$

Based on the synonymous mutation data, the likelihood for gene $g$ is:

$$P(Y^{S_g} | \beta^b, \alpha) = \int \prod_{i,t \in S_g} P(Y_{it} | \mu_{it}(\beta^b), \lambda_g) P(\lambda_g | \alpha) d\lambda_g, \tag{8}$$

which has a closed form (see Supplementary Note 4). Assuming independence across genes yields the likelihood for synonymous mutations:

$$L^S(\beta^b, \alpha) := \prod_g P(Y^{S_g} | \beta^b, \alpha). \tag{9}$$

We maximize this likelihood, using numerical optimization, to obtain estimates $\hat{\beta}^b, \hat{\alpha}$ for $\beta^b, \alpha$. By ignoring the non-synonymous mutation data when fitting the BMM, we may lose some efficiency in principle, but we gain considerable simplification in practice.

*SMM.* We next estimate the model-specific SMM parameters $\beta^{f,m}$, with the estimated BMM parameters fixed. During this step, we ignore the HMM model (i.e., we set $\theta_i^m = 1$), motivated by the fact that spatially clustered mutations are relatively rare, and so should not significantly impact the estimates of $\beta^{f,m}$.

For $m = \mathrm{OG}$, we estimate $\beta^{f,m}$ using the non-synonymous mutation data from a curated list $G_{\mathrm{OG}}$ of 53 OGs. Estimation for $\beta^{f,\mathrm{TSG}}$ is identical, except that we replace this list with a curated list $G_{\mathrm{TSG}}$ of 71 TSGs. We used the remaining genes to train the model $H_0$, as the vast majority of them should not be driver genes (the result that estimated effect sizes for $H_0$ shown in Fig. 2c, bottom panel are all close to 0 s, is consistent with this claim). Let $G_m$ denotes these sets of training genes. Let $Y^{NS_g}$ denotes the counts of non-synonymous mutations in gene $g$.

Assuming independence across genes, the likelihood for $\beta^{f,m}$ is:

$$L(\beta^{f,m}) = \prod_{g \in G_m} P(Y^{NS_g}, Y^{S_g} | \beta^{f,m}) \propto \prod_{g \in G_m} P(Y^{NS_g} | \beta^{f,m}, Y^{S_g}) \tag{10}$$

where the second line follows because $P(Y^{S_g} | \beta^{f,m})$ does not depend on $\beta^{f,m}$. The term in this likelihood for gene $g$ is given by:

$$P(Y^{NS_g} | \beta^{f,m}, Y^{S_g}) = \int \prod_{i,t \in NS_g} P(Y_{it} | \mu_{it}(\hat{\beta}^b), \gamma_{it}(\beta^{f,m}), \lambda_g) P(\lambda_g | Y^{S_g}, \hat{\alpha}) d\lambda_g. \tag{11}$$

It can be shown that

$$\lambda_g | Y^{S_g}, \hat{\alpha} \sim \mathrm{Gamma}\left( \hat{\alpha} + y_+^{S_g}, \hat{\alpha} + \mu_+^{S_g} \right), \tag{12}$$

where $\mu_+^{S_g}$ and $y_+^{S_g}$ are, respectively, the expected (considering only mutational features) and observed the number of synonymous mutations in gene $g$ (see Supplementary Note 4). The conditional mean of this distribution is $\frac{\hat{\alpha} + y_+^{S_g}}{\hat{\alpha} + \mu_+^{S_g}}$, so if $y_+^{S_g} > \mu_+^{S_g}$, then $E(\lambda_g | Y^{S_g}, \hat{\alpha}) > 1$.

We obtained the MLE of $\beta^{f,m}$ by maximizing the likelihood (Eq. (10)) numerically, and obtain the corresponding estimated standard errors using the curvature of the likelihood (see Supplementary Note 4). In tumor types with low mutation rates or sample sizes, these standard errors can be relatively large, so we borrow information from other tumor types to "stabilize" these estimates. Specifically, we use the adaptive shrinkage method[20] to "shrink" estimated values of $\beta^{f,m}$ in each tumor type toward the mean across all tumor types. This shrinkage effect is strongest for tumor types with large standard errors (Supplementary Fig. 8).

*HMM parameters.* Having estimated $\beta^b, \alpha$, and $\beta^{f,m}$, we fix their values and estimate the HMM parameters $\Theta^m$. The likelihood function involves marginalization of the hidden states of the Markov chain, which can be performed efficiently using standard methods for HMMs. We estimate $\Theta^m$ by maximizing this likelihood numerically. See Supplementary Note 4 for details.

**Gene classification.** Having estimated the model parameters as above, for each gene $g$, we compute its Bayes factor (BF) for being a driver gene as:

$$\mathrm{BF} := \frac{0.5 P(Y^{NS_g}, Y^{S_g} | H_{\mathrm{OG}}) + 0.5 P(Y^{NS_g}, Y^{S_g} | H_{\mathrm{TSG}})}{P(Y^{NS_g}, Y^{S_g} | H_0)}. \tag{13}$$

The equal weights in the numerator of this BF assume that OGs and TSGs are equally common.

This BF simplifies to

$$\mathrm{BF} = \frac{0.5 P(Y^{NS_g} | Y^{S_g}, H_{\mathrm{OG}}) + 0.5 P(Y^{NS_g} | Y^{S_g}, H_{\mathrm{TSG}})}{P(Y^{NS_g} | Y^{S_g}, H_0)}, \tag{14}$$

because $P(Y^{S_g} | H_m)$ is the same for every $m$. Computing the terms $P(Y^{NS_g} | Y^{S_g}, H_m)$ is performed using (Eq. (11)) above, substituting the estimated model parameters for each model $m$ (see Supplementary Note 5).

After obtaining the BFs, we can compute the posterior probability of being a driver gene (either OG or TSG) for every gene, and estimate the Bayesian FDR[44] for any given BF threshold. This step requires estimation of the proportion of driver genes, which we do by maximum likelihood (see Supplementary Note 5).

**Simulations.** For power analysis shown in Fig. 3a, we randomly picked a gene (*ERBB3*) and for a given number of samples, we simulated mutations under positive selection and assessed the power of detecting this gene as positively selected using different methods. We simulate background mutations using the BMM from dNdScv, which models mutation count data with negative binomial distribution given mutation rates. To account for tri-nucleotide context and nucleotide change type, dNdScv defined 192 mutation rate parameters. We use parameters estimated by dNdScv when applied to the LUSC cohort from TCGA and simulated background mutations according to their BMM. We generated mutation hotspots using a Markov chain, with hotspot frequency $10^{-5}$, and average length 5 bp. We then simulate positively selected non-synonymous mutations with mutation rates three times higher than the background mutation rate in non-hotspot sites, and 3000 times higher than the background rate in hotspot sites. This simulation procedure was performed 3000 times, and each time we obtained a $p$-value for each method. Power is defined as the fraction of simulations with significant $p$-values ($p < 0.05$). For the "dN/dS" method, we implement a simple test that resembles the dN/dS test. We compute the test statistics for this method, $T = \frac{\mathrm{Poisson}(y^{NS}; \mu^{NS} \gamma)}{\mathrm{Poisson}(y^{NS}; \mu^{NS})}$, where $y^{NS}$ is the total number of non-synonymous mutations in the gene, $\mu^{NS}$ is the total background mutation rate of non-synonymous mutations, and we fix $\gamma = 3$ (the true value used in simulation). The test statistics for the "cluster" method is the maximum number of mutations within 3-bp windows normalized by overall mutation rates. Null distributions of test statistics are obtained by simulating 5000 null data sets with mutation rates equal to the BMRs. The $p$-value for the "combined" method is obtained by using Fisher's method to combine the $p$-values from the "dN/dS" and "cluster" methods.

For the simulations performed in Fig. 3b–d, we selected 324 genes to be TSGs or OGs. These 324 genes included the 71 "known" TSGs and 53 "known" OGs from our training sets, plus 200 additional randomly selected genes (120 of which were randomly designated as TSGs, leaving the other 80 as OGs). All remaining genes were designated non-cancer related ($H_0$). We used the BMM from dNdScv as for the simulations for Fig. 3a (described above). For the SMM, we used the parameter values obtained from applying driverMAPS to the LUSC cohort from TCGA, but to incorporate model misspecification we added an independently simulated random effect $\epsilon_{it} \sim \mathrm{Normal}(0, .0.2^2)$ to the strength of selection at each position in each of the 324 non-null genes. To further incorporate model misspecification, when running driverMAPS we used only three of the five functional covariates (LoF, conservation, MutationAssessor) included in the SMM. We ran driverMAPS with the 71 known TSGs and 53 known OGs as the training set, and these genes were excluded when computing the ROC curve to ensure fair comparisons.

**Comparison of gene prediction results from different methods.** When comparing methods, we used the same mutation data (after filtering) and the same nominal FDR threshold (0.1) for each method. Because driverMAPS uses 124 known cancer genes as a training set, to avoid bias towards this subset of genes when computing precision or power for driverMAPS, we ran driverMAPS using a leave-one-gene-out strategy. We perform 124 runs, each time omitting one TSG/OG from the training set and estimating model parameters from the remaining genes, and then count the omitted gene as "significant" only if this gene is significant (FDR < 0.1) in this run. We then calculate precision as the percentage of significant known cancer genes of all significant genes. All results related to driverMAPS (basic, + feature and full version) presented in Fig. 4 were obtained in this way. (In fact, estimated model parameters are quite stable across runs, and so the overall result is similar to a single run not using this "leave-one-gene-out" strategy.)

**Validation of novel significantly mutated genes using expression and copy number variations data.** We downloaded RNA sequencing and copy number variations (CNV) data for the 20 tumor types from cBioPortal[45,46] (date accessed: 2017 April). For each gene, we averaged RNA Seq V2 RSEM for all individuals in

each cohort as the expression level. We also counted copy number gain and loss frequency for each gene in each cohort. To test if the novel genes are more highly expressed than non-cancer genes, we extracted the expression level of each novel gene in the tumor type that it was identified from and took median of all novel genes. Then, we randomly selected non-cancer genes matching the number of novel genes identified in different tumor types took median of tumor type-specific expression level. We repeated random selection 10,000 times, and assessed how many of these selections had median expression more than the group of novel genes to get a *p*-value. Similarly, we assessed if CNV gain and loss frequencies are higher for novel genes and we get *p*-values for novel TSGs and OGs separately.

**Gene-set enrichment analysis**. We used geneSCF to perform GO-term enrichment analysis for novel genes[33]. For GO-molecular function (MF) terms, we identified 20 terms with FDR < 0.1. We presented the significant gene set in Fig. 5f. There are a few gene set associated with multiple GO terms, and we selected one of them. For GO-cellular component (CC) and GO-Biological process (BP) terms, most significant gene set were also captured by GO-MF, so we did not show it in main figures. For GO-BP, we identified one significant term:

- GO0017196~N-terminal peptidyl-methionine acetylation (genes: NAA30; NAA25; NAA16).

  For GO-CC, we identified five significant terms:

- GO:0005720~nuclear heterochromatin (genes: HIST1H1E; HIST1H1B; EME1)
- GO:0036396~RNA N6-methyladenosine methyltransferase complex (genes: METTL3; METTL14)
- GO:0045252~oxoglutarate dehydrogenase complex (genes: OGDH; OGDHL)
- GO:0048476~Holliday junction resolvase complex (genes: RAD51C; EME1)
- GO:0030868~smooth endoplasmic reticulum membrane (genes: HSD3B2; TTYH1).

**Cell lines, siRNA knockdown, and plasmid transfection**. The T24 cells used in functional validation of METTL3 were purchased from ATCC (HTB-4) and grown in McCoy's 5A medium (Gibco, 16600) supplemented with 10% FBS (Gibco), and 1% penicillin–streptomycin (Gibco, 15140). The 5637 cells were purchased from ATCC (HTB-9) and grown in the RPMI-1640 medium (Gibco, 11875) supplemented with 10% FBS and 1% penicillin–streptomycin. Both cell lines were not tested for mycoplasma contamination and authenticated. Construction of the pcDNA3 plasmids for the expression of mutated METTL3 in mammalian cells was achieved by using a Q5® Site-Directed Mutagenesis Kit (NEB) following the manufacturer's protocols.

Sequencing primer used are provided in Supplementary Note 7. All siRNAs were ordered from QIAGEN. All stars negative control siRNA (1027281) was used as a siRNA control. Sequences for METTL3 is 5′-CGTCAGTATCTTGGGCAAG TT-3′. Transfection was achieved by using Lipofectamine RNAiMAX (Invitrogen) for siRNA, or Lipofectamine 2000 (Invitrogen) for the plasmids following manufacturer's protocols.

**In vitro assay for m⁶A methyltransferase activity**. The recombinant, His-tagged proteins METTL14 with wild-type or mutant METTL3 were expressed in 1 LB *E. coli* expression system, harvested, and sonicated in lysis buffer (50 mM Bis-Tris [pH 7.0], 1 M NaCl, and 1 mM DTT, and supplemented with protease inhibitors). The supernatant was loaded onto an Ni-NTA affinity column (QIAGEN), and the beads were washed with wash buffer (50 mM Bis-Tris [pH 7.0], 1 M NaCl, 1 mM DTT, and 20 mM Imidazole [pH 7.0]) and eluted with elution buffer (20 mM Bis-Tris [pH 7.0], 1 M NaCl, 1 mM DTT, and 250 mM Imidazole [pH 7.0]). Target proteins were further purified by ion-exchange chromatography. Protein purity was assessed by SDS-PAGE, and protein concentration was determined by UV absorbance at 280 nm. We performed an in vitro methyltransferase activity assay in a 50 μL of reaction mixture containing the following components: 0.15 nmol RNA probe, 0.15 nmol each fresh recombinant protein (METTL14 combination with an equimolar ratio of METTL3 or mutant METTL3), 0.8 mM $d3$-SAM, 80 mM KCl, 1.5 mM MgCl₂, 0.2 U μL-1 RNasin, 10 mM DTT, 4% glycerol, and 15 mM HEPES (pH 7.9). The reaction was incubated for 12 h at 16 °C, RNA was recovered by phenol/chloroform (low pH) extraction followed by ethanol precipitation, and was digested by nuclease P1 and alkaline phosphatase for LC-MS/MS detection. The nucleosides were quantified by using the nucleoside-to-base ion mass transitions of 285 to 153 ($d3$-m⁶A) and 284 to 152 (G).

**RNA isolation**. The total RNA was isolated with TRIZOL reagent (Invitrogen). mRNA was extracted from the total RNA using the Dynabeads® mRNA Purification Kit (Invitrogen), followed by removal of contaminating rRNA with the RiboMinus transcriptome isolation kit (Invitrogen). mRNA concentration was measured by UV absorbance at 260 nm.

**LC-MS/MS quantification of m⁶A in poly(A)-mRNA**. In all, 100–200 ng of mRNA was digested by nuclease P1 (2 U) in 25 μL of buffer containing 25 mM of NaCl, and 2.5 mM of ZnCl₂ at 42 °C for 2 h, followed by the addition of NH₄HCO₃

(1 M, 3 μL) and alkaline phosphatase (0.5 U) and incubation at 37 °C for 2 h. The sample was then filtered (0.22 -m pore size, 4 -mm diameter, Millipore), and 5 μL of the solution was injected into the LC-MS/MS. The nucleosides were separated by reverse phase ultra-performance liquid chromatography on a C18 column with online mass spectrometry detection using Agilent 6410 QQQ triple-quadrupole LC mass spectrometer in positive electrospray ionization mode. The nucleosides were quantified by using the nucleoside to base ion mass transitions of 282 to 150 (m⁶A), and 268 to 136 (A). Quantification was performed by comparison with a standard curve obtained from pure nucleoside standards run with the same batch of samples. The ratio of m⁶A to A was calculated based on the calibrated concentrations.

**Cell-proliferation assay**. Five thousand cells were seeded per well in a 96-well plate. The cell proliferation was assessed by assaying the cells at various time points using the CellTiter 96® Aqueous One Solution Cell Proliferation Assay (Promega) following the manufacturer's protocols. For each cell line tested, the signal from the MTS assay was normalized to the value observed ~24 h after seeding.

**Reporting summary**. Further information on research design is available in the Nature Research Reporting Summary linked to this article.

## Data availability
The simulated dataset may be downloaded from Zenodo [https://doi.org/10.5281/ zenodo.2932987], and the filtered somatic mutation lists from 20 tumor types that were used as input files for driverMAPS may also be downloaded from Zenodo [https://doi. org/10.5281/zenodo.1209412]. Original somatic mutation list files without filtering were downloaded from TCGA GDAC website (version: analyses__2016_01_28) [https://gdac. broadinstitute.org/]. RNA sequencing and CNVs data for the 20 tumor types were downloaded from cBioPortal [https://www.cbioportal.org/]. Gene annotation data were downloaded from GENCODE (version 19, Feb 2014) [https://www.gencodegenes.org/]. The source data underlying Figs. 6C, 6D, Supplementary Figs. 6 and 7 are provided as a Source Data file.

## Code availability
The driverMAPS software is available from the driverMAPS website [https://szhao06. bitbucket.io/driverMAPS-documentation/docs/download.html]. The source code for driverMAPS is available from the Bitbucket repository [https://bitbucket.org/szhao06/ maps]. Other software used in this study are TCGA GDAC firehose_get (version 0.4.6) [http://gdac.broadinstitute.org/] and ANNOVAR (version 2016Feb01) [http://annovar. openbioinformatics.org/en/latest/].

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

## Acknowledgements

We thank Wei Du for helpful discussion and Megan McNerney for critical comments of the paper. We thank Peter Carbonetto for helping with software development. We also thank members of Stephens' and He's labs for useful feedbacks during the course of this project. This work was supported by National Institutes of Health (NIH) grants MH110531 (to XH) and HG002585 (to MS).

## Author contributions

X.H., M.S. and S.Z. conceived the study. X.H., M.S. and S.Z. developed the methods. S.Z. designed the algorithms, implemented the software. S.Z., P.N., Y.L., E.C. and N.K. performed the analyses. J.L. and C.H. designed and performed experiments regarding METTL3. X.H., M.S., S.Z. and J.L. wrote the paper.

## Additional information

**Competing interests:** The authors declare no competing interests.

