## [Peer Review File · Nature Communications]

Reviewers' comments:

Reviewer #1 (Remarks to the Author):

The authors developed driverMAPS (Model-based Analysis of Positive Selection), a comprehensive model-based approach to driver gene identification. Simulation analysis showed that driverMAPS is more powerful than existing software. More interestingly, the authors identified several novel driver genes, including the mRNA methyltransferases. Their results suggested that mRNA modification is an important biological process in cancer.

Major comments:

1. Since driverMAPS is based on a much richer statistical model, could the author specify the performance, such as speed, memory, and storage to run this algorithm?
2. The authors included several algorithms to compare the performance, such as MuSiC. The authors may also include several other famous software for comparison, including CanDrA, CHASM, MutationAssessor, etc.
3. The reviewer appreciated the better performance for driverMAPS based on simulation data. It may be more convincing to evaluate the performance based on real biological data, especially as mentioned by the author, the "cancer dependency map", as well as another study (Ng et al., Systematic Functional Annotation of Somatic Mutations in Cancer, Cancer Cell, 2018). Did driverMAPS identify more driver genes overlapped with these two datasets than any other algorithms? If using these two datasets as the true functional calls, what is the AUC/true positive/false positives across different algorithms?
4. It is not convincing for figure 5b-c since it is not surprising to see the statistical significance when comparing novel significant genes with random genes, especially these genes are expressed at a higher level than non-driver genes. Did these significant genes show higher expression level/CNV alterations than gene lists identified by other algorithms?
5. The algorithm mentioned the single-base level, while it seems unnecessary to identify driver genes. Could the authors clarify? If this is really the functions of the algorithm, could the author use this directly to predict which mutation on METTL3 is a driver?

Minor comments:

1. It is difficult to interpret the results for figure 6c/d without appropriate label. The color does not match from legend to the figure.
2. Could the authors also include the paper for METTL14 in endometrial cancer (Chuan He, to appear) in the revision for reference?

Reviewer #2 (Remarks to the Author):

Zhao et al present a novel method, driverMAPS, to nominate driver genes in exome SNV data. The approach is based on a Bayesian hypothesis testing approach that classifies genes according to being an oncogene or TSG driver or null. They benchmark the method against state of the art approaches (dndscv, MutSigCV, oncodrive suite) to demonstrate increased power and adequate false discovery control with driverMAPS. They validate one of the novel hits derived from their method using functional experimentation.

Overall, the method is conceptually sound, clearly presented (though with numerous typos), and assiduously benchmarked against state of the art approaches. Functional mutations in an RNA

methyltransferase (METTL3) previously not associated with cancer are validated in cell lines using RNAi and transgene over-expression in combination with an RNA methylation readout.

DriverMAPS employs a base resolution modeling of background mutation rates and explicit modeling of alternative hypotheses (oncogene, tumor suppressor) to derive a Bayes Factor of a gene being a cancer driver. The background likelihood models context-specific mutation counts at each position of the exome as an overdispersed Poisson whose log mean is a linear combination (defined by "Beta_b" coefficients) of gene level log mutation rate and regional covariates (gene expression, hi-c). The "selection model" likelihood adds additional "functional" terms to this log mean parameter. These comprise "Beta_f" coefficients that specify linear combinations of "functional features", e.g. SIFT, PhyloP, as well as an HMM derived "hotspot term" theta.

Model fitting involves finding maximum likelihood assignments for these Beta coefficients and the theta coefficients, among others, against real data of mutation counts. Background models are fit genome-wide against synonymous counts only. The two separate selection models (oncogene and tumor suppressor) are fit against non-synonymous counts using small curated sets of (<100) oncogenes and tumor suppressor genes. The resulting models are then applied genome-wide to yield Bayes factors for each genes, which are then combined using a Bayesian FDR procedure to yield a list of driver genes for the given tumor type.

Major critiques:

- Justify / Clarify model fitting and necessity of curated "training sets" of oncogenes and tumor suppressors

The selection model is fit to curated tumor suppressors and oncogenes - this reliance makes the method seem a little flimsy and circular, since many of these "lists" have been derived from similar analyses (eg MutSig) applied to TCGA

These genes are presumably chosen from a "pan-cancer" list but the models fit by tumor type, where the majority of these genes are not relevant - e.g. the majority of EGFR mutations in melanoma are passengers. This makes me wonder how essential this gene choice is for the model fitting. Indeed the correlation of TSG with CNV loss frequency (Figure 4) is poor, so not clear how much additional signal the TSG / OG training is picking up.

How well would a single "driver" model trained on all genes perform compared to this model? Similarly, how well does this OG + TSG driver model perform as an OG / TSG "classifier" on cross validation eg if you leave half of the genes out of the training? My guess it's not spectacular.

The background model / B^b parameters is fit using synonymous mutations only. Are these B^b parameters used in the selection model or re-fitted in the selection model?

What happens when nonsynonymous mutations are used instead? Since the background model lacks functional and hotspot features, it should still show a difference vs the selection model.

B^f parameters (SIFT, CONS, PhyloP) are reported in Figure 2c for the background model (B^f_0) - but the BMM definition does not include a selection term so not clear how these are fit. eg using the "selection model" on synonymous data?

- Please describe and justify simulation approach used for power analysis

The simulations are core to the arguments that driverMAPS has increased power and adequate FDR control, however they are not rigorously specified. "We simulated mutations under positive selection" and "we simulate synonymous mutations at predefined background mutation rates" are quite vague. One guess is that the authors are using the inference model with some specified

parameters as a generative model from which to draw mutation count data. If so, then the simulations seem somewhat "rigged" to favor driverMAPS. This is especially true if the generative model has the exact same structure (eg same set of background and functional covariates) as the inference. What would happen if you had 10 unknown functional covariates generating the positively selected data, or only considered a subset of the generative functional / background covariates in the driverMAPS inference. ie how does incomplete knowledge of these factors influence the power and precision of the inference.

It would be ideal if the authors could use a more objective benchmark e.g. a "third party" cancer mutation simulation software or analysis of subsampled "real" data. There is no third party software that I'm aware of, and analysis of subsampled data vs a gold standard (eg COSMIC) may provide a decent analysis of specificity but sensitivity is hard to quantify. However, the authors should either pursue something in this direction or at the very least present a rigorous description of the simulations used for benchmarking.

Minor critiques:

- Please clarify the HMM model and its inference, especially the theta term

I've read the main and supplementary methods. It's not clear what theta represents, since it depends on p_m , which is not explicitly defined in the supplement via a formula but only described as the "average increase of mutation rate in hotspots under model m". Conceptually theta should be a variable whose value is >1 at hotspots and is 1 otherwise ... eg like a relative risk for the binary variable of "is hotspot" vs "is not hotspot".

Please provide an explicit formula for theta and ρ_m . There is also a parameter ρ_0 , and ρ mentioned in the HMM model fitting (page 7 supplement) that is not defined in the HMM spec (page 6). My guess is that ρ is some odds ratio of being a hotspot, but this is not clear.

The initial HMM parameters (p_0 , p_1) btw are strangely defined - these params don't have a subscript i but yet they are defined in terms of Z_i . Are these just supposed to be the probabilities of $P(Z_0^m = 1)$.

- Spellcheck!

There are a bunch of typos in this manuscript and the supplement. The following is by no means comprehensive: Figure 3 "false postive" . Supplement page 7: "The mission probability", "emission prababilities", Supplement page 5: "values of t are limit to the..."

Response to reviewer 1:

Major comments:

1. Since driverMAPS is based on a much richer statistical model, could the author specify the performance, such as speed, memory, and storage to run this algorithm?

We have provided a website for detailed installation, usage and performance information. Please see <https://szhao06.bitbucket.io/driverMAPS-documentation/docs/index.html>. The software and output files need a total of 8G storage. We have dramatically improved performance of the software since the previous version. Currently, driverMAPS v1.0.4 can be run using a single CPU processor, with 12G memory. The single processor mode takes around 10 hours to finish. To increase speed, we have provided a parallel computing option and the run time can be reduced to 2-3 hours when using 6 cores and 18G memory in total. We have added this information to Supplementary Notes.

2. The authors included several algorithms to compare the performance, such as MuSiC. The authors may also include several other famous software for comparison, including CanDrA, CHASM, MutationAssessor, etc.

We did not compare driverMAPS with CanDrA, CHASM and MutationAssessor, because these software packages all address a different problem to the one addressed by driverMAPS (and the other software we did compare against). Specifically, CanDrA, CHASM and MutationAssessor typically use a training set to learn whether *an individual mutation* is likely a cancer driver mutation based on features of the mutation, e.g. conservation, impact on protein structure. This type of method, alone, is not best suited for discovering cancer driver genes. For example, a mutation in a conserved position at a kinase domain may be statistically more likely to be a driver mutation than random mutations, but this will not automatically make the gene containing this mutation a driver gene. One needs to integrate information of the entire gene to assess its role - which is what driverMAPS does (as do the methods we compare against). Because of these different goals, the softwares mentioned by the reviewer complement, instead of competing with, our method. Indeed, we used scores from MutationAssessor as a functional annotation in the driverMAPS model.

Since these software packages address a different problem we have not changed the manuscript in light of this comment.

3. The reviewer appreciated the better performance for driverMAPS based on simulation data. It may be more convincing to evaluate the performance based on real biological data, especially as mentioned by the author, the "cancer dependency map", as well as another study (Ng et al., Systematic Functional Annotation of Somatic Mutations in Cancer, Cancer Cell, 2018). Did driverMAPS identify more driver genes overlapped with these two datasets than any other algorithms? If using these two datasets as the true functional calls, what is the AUC/true positive/false positives across different algorithms?

We agree with the reviewer that there are limitations to our evaluation of methods, although we emphasize that we tried to make our simulations as realistic as possible by ensuring that they capture the main biological patterns exhibited by real cancer driver genes (functional bias of mutations, spatial clustering, etc.). In response to comments made by another reviewer we have further improved the simulations to allow for model misspecification - specifically we allow the mutation effects to vary significantly from what is expected based on functional annotations. Additionally, we used a more sophisticated (and realistic) background mutation model in our new simulations. The new results show that the superior performance of driverMAPS is robust to these changes. Please see our updated simulation procedures in the Method section and results from these new simulations in Figure 3 in the manuscript.

Regarding the real data, the difficulty is that there is no “true set” of driver genes for us to perform evaluation. While the two datasets mentioned by the reviewer are definitely relevant to our study, each of them has limitations that makes them difficult to use to confirm our results. The genes provided by the cancer dependency map paper are not necessarily cancer driver genes. In the paper [Tsherniak and Hahn, Cell, 2017], the authors reported a total of 769 genes displaying strong “differential dependency” in about 500 cancer cell lines (the genes that are universally required are not interesting). The authors partition these genes into several classes, with only class I (47 genes, less than 10%) representing likely oncogenes. The rest are various genes that cancer cells’ growth depends on. The second paper, Ng et al, Cancer Cell, 2018, performed experimental studies on mutations in a very small set of well-known cancer driver genes (half of mutations tested come from just 4 driver genes!). As such, the dataset has limited value in assessing methods that aim to predict novel driver genes.

Our approach was to compile a relatively reliable and comprehensive set of currently known driver genes. It is composed of COSMIC census gene list (manually curated with experimental support, <https://cancer.sanger.ac.uk/census>), TCGA pan-cancer gene list (driver genes discovered by TCGA project) and Vogelstein et al 2013 gene list (manually curated with experimental support). Using this as the “true set”, we found driverMAPS recovered the largest number of known driver genes compared to software with similar fractions of known driver genes out of all significant gene (Figure 4 a, b c). We note that even this list is not complete and this is exactly the reason why we are developing methods for discovering novel driver genes.

4. It is not convincing for figure 5b-c since it is not surprising to see the statistical significance when comparing novel significant genes with random genes, especially these genes are expressed at a higher level than non-driver genes. Did these significant genes show higher expression level/CNV alterations than gene lists identified by other algorithms?

The analysis performed in Figure 5b-c simply aims to assess the novel genes discovered by driverMAPS using independent genomic data. If they are enriched with driver genes, we should see elevated expression and higher chance of concurrence with CNV events. Comparisons with other algorithm in terms of sensitivity, specific *etc*, are already performed in a more accurate way as shown in Figures 3 and 4.

5. The algorithm mentioned the single-base level, while it seems unnecessary to identify driver genes. Could the authors clarify? If this is really the functions of the algorithm, could the author use this directly to predict which mutation on METTL3 is a driver?

To clarify, this is not the function of the algorithm. Although the *model* is at the single-base level (i.e., every different possible mutation has its own rate depending on its unique set of features) the function of our algorithm is to **combine information** across all mutations to give gene level predictions, i.e. whether a gene is a driver or not. The reason the model is performed at the single-base level is to *model the heterogeneity of background mutation rates and functional features across the gene* and not to make predictions for each mutation.

However, the reviewer raised an interesting question, that is, besides giving gene-level predictions, can we prioritize positions as likely driver mutations? Taking METTL3 as an example, there are 17 different nonsynonymous mutations identified in the bladder cancer cohort and was identified as a significant tumor suppressor gene (TSG). We rank these mutations by their contributions to the evidence of the gene as TSG. Specifically, we calculated a Bayes factor for each mutation, which compares the likelihood under the selection model against the background model. We provide the results below, and we do feel that they provide some helpful indications of which mutations are contributing the most to drive the gene-level signal. We therefore added these results to the manuscript (Supplementary Table S10).

Position (on Chr14)	Mutation	log(Bayes Factor)	Amino acid change	Loss of function	Conser- vation	Sift_ pred	Phylop_ _pred	MA_ _pred	No. mutations
21967254	C>T	2.95	E516K	No	Yes	Yes	Yes	Yes	2
21971663	G>A	2.37	Q126X	Yes	Yes	Yes	Yes	No	1
21971651	C>A	2.17	E130X	Yes	Yes	No	Yes	No	1
21969218	T>A	1.48	N318I	No	Yes	Yes	Yes	Yes	1
21967257	C>A	1.48	D515Y	No	Yes	Yes	Yes	Yes	1
21967260	G>T	1.48	P514T	No	Yes	Yes	Yes	Yes	1
21967206	C>G	1.47	E532Q	No	Yes	Yes	Yes	Yes	1
21969223	G>C	1.47	F316L	No	Yes	Yes	Yes	Yes	1
21967206	C>T	1.47	E532K	No	Yes	Yes	Yes	Yes	1
21967452	C>T	1.47	E506K	No	Yes	Yes	Yes	Yes	1
21967728	C>T	1.47	E454K	No	Yes	Yes	Yes	Yes	1
21967676	C>T	1.45	R471H	No	Yes	Yes	Yes	Yes	1
21967685	C>T	1.45	R468Q	No	Yes	Yes	Yes	Yes	1
21971844	G>A	0.91	P94L	No	Yes	Yes	Yes	No	1
21969985	C>T	0.69	E262K	No	Yes	No	Yes	No	1
21967264	A>C	0.59	H512Q	No	Yes	No	No	No	1
21972010	C>T	0.40	E39K	No	No	No	Yes	No	1

Minor comments:

1. It is difficult to interpret the results for figure 6c/d without appropriate label. The color does not match from legend to the figure.

We thank the reviewer for pointing out this error. We have now corrected the color labels and they should now match with the figure.

2. Could the authors also include the paper for METTL14 in endometrial cancer (Chuan He, to appear) in the revision for reference?

We thank the reviewer for this reminder. The paper has now been published and reference has been updated in the manuscript.

Reference Liu, J. et al. m6A mRNA methylation regulates AKT activity to promote the proliferation and tumorigenicity of endometrial cancer. *Nat. Cell Biol.* 20, 1074–1083 (2018).

Response to reviewer 2:

Major critiques:

- Justify / Clarify model fitting and necessity of curated "training sets" of oncogenes and tumor suppressors

The selection model is fit to curated tumor suppressors and oncogenes - this reliance makes the method seem a little flimsy and circular, since many of these "lists" have been derived from similar analyses (eg MutSig) applied to TCGA

These genes are presumably chosen from a "pan-cancer" list but the models fit by tumor type, where the majority of these genes are not relevant - e.g. the majority of EGFR mutations in melanoma are passengers. This makes me wonder how essential this gene choice is for the model fitting. Indeed the correlation of TSG with CNV loss frequency (Figure 4) is poor, so not clear how much additional signal the TSG / OG training is picking up.

We used a small training set of 124 curated tumor suppressor genes and oncogenes from Vogelstein *et al* 2013. Only 29% of these genes were uncovered by unbiased genome sequencing studies using programs such as MutSig; the rest were discovered by more direct investigations with strong experimental support. Furthermore, even those uncovered by MutSig-type analyses have been carefully curated, and generally have strong additional experimental support. Therefore, the parameters we learn from this training set should primarily reflect properties of real cancer genes rather than just re-learning parameters used by MutSig. Indeed, in support of this, the parameter estimates we obtained match biological expectations, e.g. we see mutations enriched in conserved positions, even though MutSig does not use conservation information. Further, our method does identify many potential novel driver genes, and the list of novel driver genes we identify is not simply recapitulating MutSig results, demonstrating that our method is not "circular".

Regarding the "necessity" of training data, competing analysis methods -- whether or not formally based on training data -- incorporate parameter values that reflect current understanding of the characteristics of driver genes. For example, MutSig used a "null boost score" of 3 for null mutations and score of 1 for other nonsynonymous mutations. Our method, instead of heuristically assigning values to such parameters, learns them from data, which we see as an advantage rather than a limitation.

The reviewer is correct that, due to cross-tumor differences the pan-cancer training set will likely contain false driver genes for any given tumor type. Also, the labeling of TSG or OG may not be accurate for all genes. Similarly the training set of “null” genes will inevitably include some actual true (as yet undiscovered) driver genes. It is important to note that although this “contamination” will reduce power to detect driver genes, it will not increase false positive prediction rate. This is because including non-driver genes in the training set will cause underestimation of the enrichment parameters for driver genes. Despite these limitations, the current training set should be enriched with driver genes, enabling us to capture positive selection signals. From results shown in the paper, its performance is already better than other currently available methods.

We have added a paragraph addressing these points to the manuscript (see line 107 -118).

How well would a single "driver" model trained on all genes perform compared to this model?

To assess the “single driver” model, we pooled all training genes into one set, without distinguishing OGs and TSGs, and estimated parameters for the selection model. The following table showed estimated parameter using simulated mutation data of 200 samples (the same data as used in Figure 3b-d). We see the effect sizes for functional covariates in the single driver model are a compromise between parameters estimated separately for TSG and OG model. For parameters related to the HMM, as there are less frequent hotspots when pooling TSG and OG together, we see smaller v_{01} (transition probability from non-hotspot to hotspot) for single driver model. Thus, this single driver model is also able to capture various positive selection signals.

TSG model	β^{LoF}	β^{cons}	β^{MA}	β_o^f							
	1.387	0.247	0.664	0.873							
OG model	β^{LoF}	β^{cons}	β^{MA}	β_o^f	Parameters related to HMM						
					p_1	p_2	v_{01}	v_{01}	v_{01}	v_{01}	$\log(\rho_1/\rho_0)$
	-0.189	0.548	0.337	0.194	0.999	0.000591	1.000	0.000388	0.656	0.344	8.654
Single driver model	β^{LoF}	β^{cons}	β^{MA}	β_o^f	Parameters related to HMM						
					p_1	p_2	v_{01}	v_{01}	v_{01}	v_{01}	$\log(\rho_1/\rho_0)$
	0.655	0.417	0.492	0.719	1.000	0.000194	1.000	0.000127	0.655	0.345	8.079

In fact, except for the β^{LoF} parameter, most parameters are not affected much in the single driver model vs. the separate models. Indeed, with 200 simulated samples, the difference of performance between the TSG/OG model and a single model is minimal. With growing sample sizes, TSG/OG model has only slightly higher more power. See results in the following figure.

Despite the fact that the single-driver model has similar performance, we prefer to model TSG and OGs separately, because i) biologically it makes sense that patterns of selection should be different in the two groups, and indeed our estimates of β^{LoF} correctly reflect this biological expectation; ii) in the future (e.g. with richer training data) it is possible that the gain over a single-driver model may be greater.

We have added these results and related discussion in the manuscript from line 190 to 197, and added the figure in the supplement (supplementary figure S4).

Similarly, how well does this OG + TSG driver model perform as an OG / TSG "classifier" on cross validation eg if you leave half of the genes out of the training? My guess it's not spectacular.

To assess the performance of our model as a TSG/OG classifier, we performed the cross validation analysis suggested by the reviewer. We performed five-fold cross-validation, in which 80% of the original training set is used for training, and the remaining 20% of genes are used for evaluating TSG/OG labels (validation set). We used TCGA data for 20 tumor types, and predict the TSG/OG labels for genes in the validation set in each tumor type where the genes are identified as significant (FDR <0.1). The result below show most predictions are consistent with the gene's original label in the training set (91% consistency). Looking into the genes, we found several classical TSGs or OGs could be reliably labeled. For example, PTEN is highly enriched with LoF mutations and has been correctly identified as TSG in all 13 tumor types it appeared significant; PIK3CA and KRAS mutations have strong spatial clustering pattern and were correctly predicted as OG in 24 out of 25 cases.

However, we feel these results may overstate the ability of our method as a TSG/OG classifier in practice. Specifically, we believe the results may be biased towards the rules used by Vogelstein 2013, which defined TSGs largely based on enrichment of LoF mutations and OGs based on spatial clustering. It is unclear how general these rules are applicable to other, less common driver genes; as a result, we did not include these results in the manuscript or emphasize our model as a TSG/OG classifier. We thank the reviewer for pointing out the weakness of using it as TSG/OG classifier, we think this is an important issue and have added clarifications and discussions about this in the manuscript from line 183 to 189.

The background model / B^b parameters is fit using synonymous mutations only. Are these B^b parameters used in the selection model or re-fitted in the selection model?

They are used in the selection model. We now clarify this in line 492-493 in method section of the manuscript and figure 1 legend (line 752).

What happens when nonsynonymous mutations are used instead? Since the background model lacks functional and hotspot features, it should still show a difference vs the selection model.

In Figure 2c bottom panel, we trained the SMM using all genes not included in the training sets of “known” cancer genes. Specifically, we first used synonymous mutations to estimate background model parameters, then fit the selection model using nonsynonymous mutations outside the training gene set (with the background model parameters plugged in). The Figure shows that the effect size parameters are all close to 0, which demonstrates that nonsynonymous mutations in non-cancer genes are explained well with the background model estimated from synonymous mutations. This implies that using nonsynonymous mutations to estimate background rates would yield results similar to

those we obtained using synonymous mutations. We have added clarifications in line 162-167 of the manuscript and legend of Figure 2 (line 766-768).

B^f parameters (SIFT, CONS, PhyloP) are reported in Figure 2c for the background model (B^f 0) - but the BMM definition does not include a selection term so not clear how these are fit. eg using the "selection model" on synonymous data?

As clarified above, Figure 2c bottom panel is not background model and thus not fitted using synonymous mutation data. Instead, it is the selection model (without HMM part) fitted to nonsynonymous mutations outside the training genes. We have made this clearer in Figure 2c legend.

- Please describe and justify simulation approach used for power analysis

The simulations are core to the arguments that driverMAPS has increased power and adequate FDR control, however they are not rigorously specified. "We simulated mutations under positive selection" and "we simulate s mutations at predefined background mutation rates" are quite vague. One guess is that the authors are using the inference model with some specified parameters as a generative model from which to draw mutation count data. If so, then the simulations seem somewhat "rigged" to favor driverMAPS. This is especially true if the generative model has the exact same structure (eg same set of background and functional covariates) as the inference. What would happen if you had 10 unknown functional covariates generating the positively selected data, or only considered a subset of the generative functional / background covariates in the driverMAPS inference. ie how does incomplete knowledge of these factors influence the power and precision of the inference.

It would be ideal if the authors could use a more objective benchmark e.g. a "third party" cancer mutation simulation software or analysis of subsampled "real" data. There is no third party software that I'm aware of, and analysis of subsampled data vs a gold standard (eg COSMIC) may provide a decent analysis of specificity but sensitivity is hard to quantify. However, the authors should either pursue something in this direction or at the very least present a rigorous description of the simulations used for benchmarking.

We thank the reviewer for pointing out the potential bias towards driverMAPS in our previous simulation procedures. We agree a "third party" mutation simulation would be ideal to test the performance, but as the reviewer also pointed out, no such software is available. Thus, to provide a fairer comparison, we made several changes to our simulation scheme, allowing the simulated data to deviate from our inference model.

First, we changed our background mutation simulation procedures. Instead of using our own background mutation model, we used the background mutation model of dNdScv (Martincorena, I. *et al.*2017). In dNdScv, 192 mutation types (based on tri-nucleotide context) were used; in comparison, 9 types were used in our model. We simulated data using parameters estimated by dNdScv for the LUSC cohort from TCGA (see simulation procedures in manuscript for details). Second, in our original model, the effect of a mutation (log-relative risk) is a linear function of the covariates. We now add a noise term to the linear model in the simulation (this can be thought of as random effects), so that the effect of a mutation in all simulated TSG or OG genes may deviate from the linear model. We use Normal(0, 0.2²) for the random effect term. Third, we simulate with 5 functional covariates

(LoF, conservation, SiFT, PhyloP, MutationAssessor) but only used three of them (LoF, conservation, MutationAssessor) when running driverMAPS.

With these changes, we have re-written the simulation part in the main text (line 199 to 239). We also have re-written the simulation procedure, with more details, in the Method section (line 537 to 570). Below are the new simulation results and we have replaced Figure 3 with these new results. Our method remains as the best performing software. Interestingly, we note that MutSigCV now has significantly inflated type I error rate, suggesting that it may be sensitive to misspecification of background mutation rate model.

Minor critiques:

- Please clarify the HMM model and its inference, especially the theta term

I've read the main and supplementary methods. It's not clear what theta represents, since it depends on p_m , which is not explicitly defined in the supplement via a formula but only described as the "average increase of mutation rate in hotspots under model m". Conceptually theta should be a variable whose value is >1 at hotspots and is 1 otherwise ... eg like a relative risk for the binary variable of "is hotspot" vs "is not hotspot".

Please provide an explicit formula for theta and ρ_m . There is also a parameter ρ_0 , and ρ_0 mentioned in the HMM model fitting (page 7 supplement) that is not defined in the HMM spec (page 6). My guess is that ρ is some odds ratio of being a hotspot, but this is not clear.

We have modified the supplement to clarify the HMM model specification. Essentially theta can take one of two values -- $\rho_1 > 1$ inside hotspots, and $\rho_0 < 1$ outside hotspots. (In

each case θ is multiplying the overall average rate, so θ is not exactly 1 outside hotspots, because the average includes both hotspots and non-hotspots.)

The initial HMM parameters (p 0, p 1) btw are strangely defined - these params don't have a subscript i but yet they are defined in terms of Z i. Are these just supposed to be the probabilities of $P(Z_0^m = 1)$.

Yes, they are just initial probabilities and should not depend on i. We have corrected this in the supplement.

- Spellcheck!

There are a bunch of typos in this manuscript and the supplement. The following is by no means comprehensive: Figure 3 "false postive" . Supplement page 7: "The mission probability", "emission prabablities", Supplement page 5: "values of t are limit to the..."

We thank the reviewer for pointing out these errors. We have corrected them and performed a thorough spellcheck. We have also made minor wording improvement throughout the manuscript and supplementary notes. Also, notations for all equations have been checked, with some minor changes.

REVIEWERS' COMMENTS:

Reviewer #1 (Remarks to the Author):

The authors addressed all of my concerns.

Reviewer #2 (Remarks to the Author):

The reviewers have adequately addressed all my concerns.

REVIEWERS' COMMENTS:

Reviewer #1 (Remarks to the Author):

The authors addressed all of my concerns.

--- We thank the reviewer for providing us the reviews.

Reviewer #2 (Remarks to the Author):

The reviewers have adequately addressed all my concerns.

--- We thank the reviewer for providing us the reviews.